# Quantifying the uncertainty of precipitation forecasting using probabilistic deep learning

Lei Xu[1], Nengcheng Chen[1,2], Chao Yang[1], Hongchu Yu[3], Zeqiang Chen[1]

[1]National Engineering Research Center for Geographic Information System, School of Geography and Information Engineering, China University of Geosciences (Wuhan), Wuhan 430074, China

[2]State Key Laboratory of Information Engineering in Surveying, Mapping, and Remote Sensing, Wuhan University, Wuhan 430079, China

[3]School of Navigation, Wuhan University of Technology, Wuhan 430063, China

*Correspondence to*: Chao Yang (yangchao@cug.edu.cn)

Abstract. Precipitation forecasting is an important mission in weather science. In recent years, data-driven precipitation forecasting techniques could complement numerical prediction, such as precipitation nowcasting, monthly precipitation projection and extreme precipitation event identification. In data-driven precipitation forecasting, the predictive uncertainty arises mainly from data and model uncertainties. Current deep learning forecasting methods could model the parametric uncertainty by random sampling from the parameters. However, the data uncertainty is usually ignored in the forecasting process and the derivation of predictive uncertainty is incomplete. In this study, the input data uncertainty, target data uncertainty and model uncertainty are jointly modeled in a deep learning precipitation forecasting framework to estimate the predictive uncertainty. Specifically, the data uncertainty is estimated a priori and the input uncertainty is propagated forward through model weights according to the law of error propagation. The model uncertainty is considered by sampling from the parameters and is coupled with input and target data uncertainties in the objective function during the training process. Finally, the predictive uncertainty is produced by propagating the input uncertainty in the testing process. The experimental results indicate that the proposed joint uncertainty modeling framework for precipitation forecasting exhibits better forecasting accuracy (improve RMSE by 1-2% and R-square by 1-7% on average) relative to several existing methods, and could reduce the predictive uncertainty by ~28% relative to Loquercio et al. (2020)'s approach. The incorporation of data uncertainty in the objective function changes the distributions of model weights of the forecasting model and the proposed method can slightly smooth the model weights, leading to the reduction of predictive uncertainty relative to Loquercio et al. (2020)'s method. The predictive accuracy is improved in the proposed method by incorporating the target data uncertainty and reducing the forecasting error of extreme precipitation. The developed joint uncertainty modeling method can be regarded as a general uncertainty modeling approach to estimate predictive uncertainty from data and model in forecasting applications.

## 1 Introduction

Precipitation is a key hydrometeorological variable in earth system science, and is the main driving factor of floods and

droughts (Xu et al., 2019). In the year of 2019, the flood disaster driven by extreme precipitation caused a direct economic loss of 29.6 billion dollars in China, and the drought disaster led to a crop production loss of 23.6 billion kilograms (http://www.mwr.gov.cn/sj/#tjgb). Accurate precipitation forecasting is vital for the early warning of flood and drought, smart city management and agricultural water resources allocation (Van Den Hurk et al., 2012; Pozzi et al., 2013). However, the precipitation forecasting problem suffers from uncertainties from data, algorithms and random factors (Reeves et al., 2014; Kobold and Sušelj, 2005; Xu et al., 2020b). The predictive uncertainty is a measurement of the spread of precipitation forecasting and could indicate how much the forecasted precipitation values fluctuate around the mean (Papacharalampous et al., 2020). Therefore, the uncertainty range should be given when generating precipitation forecasting results.

The precipitation forecasting methods can be divided into two categories: numerical weather forecasting and statistical machine learning. Numerical models consider the physical process of earth system and could simulate the interactions between atmospheres, oceans and lands (Sikder and Hossain, 2016; Molinari and Dudek, 1992). Numerical models have strong physical meaning and are the dominant ways of operational precipitation forecasting. However, the forecasting ability of numerical models is limited due to the uncertainty in initial and boundary conditions, the imperfection of parameterization schemes and the uncertainty in parameters (Reeves et al., 2014; Xu et al., 2021a). With the development of computer technology and machine learning algorithms, using random data-driven techniques for precipitation forecasting is becoming popular in recent years (Shi et al., 2015; Trebing et al., 2021; Sønderby et al., 2020). The accuracy of data-driven methods is comparable to currently advanced numerical models in short-term (e.g. from hours to weeks) precipitation forecasting. For example, the convolutional long-short term memory network is shown to outperform the physical optical flow method in precipitation nowcasting based on radar images (Shi et al., 2015). Another deep learning model called MetNet showed advantages over traditional numerical models in terms of the forecasting accuracy and running time for hourly precipitation prediction (Sønderby et al., 2020). The data-driven methods also exhibit appealing results in subseasonal to seasonal precipitation forecasting relative to numerical models (Boukabara et al., 2019; Chantry et al., 2021; Hwang et al., 2019). A key drawback of data-driven precipitation forecasting method is the lack of physical meaning, also known as black-box model. Despite this feature, data-driven statistical machine learning methods have been widely used for parameter calibration, data processing, submodel replacement and process understanding among physical simulations (Ardabili et al., 2019; Sahoo et al., 2017; Reichstein et al., 2019). The data-driven learning techniques are strong complements to numerical models for the improvement of precipitation forecasting accuracy.

The predictive uncertainty in precipitation forecasting arises mainly from data and models (Gal, 2016). The data uncertainty comes from external observation conditions, instruments and processing algorithms. The data uncertainty is usually examined by perturbing initial conditions in numerical models and producing a perturbed multi-model ensemble, which is widely seen in hydrometeorological ensemble forecasting (Xu et al., 2019; Gneiting and Raftery, 2005; Duan et al., 2019;

Vitart et al., 2017). The data uncertainty is rarely investigated in data-driven precipitation forecasting and is often assumed to be accurate without error. The model uncertainty is often represented by an ensemble of perturbed model physics and parameters in numerical weather forecasting (Vitart et al., 2017; Kirtman et al., 2014; Taylor et al., 2012). In data-driven models, the model uncertainty is generally modeled by random regularization of parameters (Gal, 2016; Kendall and Gal, 2017). For linear regression, the parametric uncertainty is indicated by the standard deviation of trained parameters. In deep learning, the network layers could be randomly abandoned to prevent overfitting and generate a forecasted ensemble by Monte Carlo sampling (Kendall and Gal, 2017; Srivastava et al., 2014; Loquercio et al., 2020; Ghahramani, 2015).

The data and model uncertainties should be considered jointly in an integrated modeling framework to get the predictive uncertainty, as the data and model uncertainties could both inflate the predictive spread considerably (Gal, 2016; Kendall and Gal, 2017). It is expected that, the forecasting result would be more or less different if the used data and parameters are randomly sampled from the population. Data uncertainty is usually assumed as a constant or Gaussian distribution and could be propagated into final forecasting through error forward propagation (Loquercio et al., 2020; Xu et al., 2020a). If the data uncertainty is unknown, it can be learned from the training process by considering the data uncertainty as a trainable parameter (Kendall and Gal, 2017). However, the joint learning of data errors and model weights will increase the number of training parameters and may mix the error flow from data and parameters. A prior estimation of data uncertainty could help unravel the data error and facilitate the training process. On the other hand, previous forecasting studies usually model the input data uncertainty and ignore the uncertainty in the target (predictand) data (Kendall and Gal, 2017; Loquercio et al., 2020). The uncertainty in the target dataset also plays an important role in the parameter training process and could influence the forecasting accuracy.

There are two ways to estimate the data uncertainty. One is to use in-situ ground stations to calculate the systematic and random errors within the data and the other is to use multisource datasets to compute random error by intercomparison (Xu et al., 2021b; Gruber et al., 2016; Sun et al., 2018). The in-situ validation method is limited to the number and density of ground stations and are suitable for small areas with enough station coverage. The second method is independent of the in-situ stations and requires multisource datasets with independent error distribution (Gruber et al., 2016). There are numerous precipitation datasets from various sensors and models and could be used to calculate precipitation data error at a large spatial scale (Xu et al., 2020b; Sun et al., 2018). Three-cornered hat (TCH) and triple collocation (TC) are two commonly used methods to evaluate the random error among multisource datasets, which do not require ground measurements as references (Premoli and Tavella, 1993; Mccoll et al., 2014; Stoffelen, 1998). The basic assumption of TCH and TC methods is the stationarity of both the raw dataset and its error, which may not be always satisfied for real-world data. Most of the existing studies assume that the used multisource datasets obey the stationarity condition when using TCH or TC methods (Xu et al., 2020b; Gruber et al., 2016; Gruber et al., 2017), which is useful for the determination of relative prior random error.

In this study, we aim to quantify the predictive uncertainty of data-driven precipitation forecasting by fully considering
the uncertainty from data and models. The data uncertainty is estimated by the TCH method a priori and is assumed as Gaussian
distribution. The data uncertainty is propagated within model training by the law of error propagation. The parametric
uncertainty is modeled by randomly abandoning some network layers during the training process. The data and model
uncertainties are jointly considered in the objective function within a deep learning encoder-decoder framework. The
forecasting experiments are conducted to see whether the accuracy of precipitation forecasting can be improved by joint data-
model uncertainty modeling relative to several uncertainty processing strategies from the existing studies.
**2 Study area and data**
The study area is located at southern and northern China, East Asia (Figure 1). The annual rainfall decreases from the
southeast to the northwest, with an approximately average rainfall of 1500 mm in the southeast regions and 300 mm in the
northwest areas. Most of the southern areas feature a subtropical monsoon climate and the rainfall is relatively larger in summer
and smaller in winter. From June to July in 2020, extreme precipitation hit the southern China (Wei et al., 2020) and caused a
direct economic loss of 13.2 billion dollars. The precipitation forecasting in southern area of China is very challenging and
meaningful. Previous studies use numerical models for precipitation forecasting in this area and show some values (Yuan et
al., 2012; Luo et al., 2017). The northern area of China features the temperate moon and continental climates, with an annual
rainfall of 400 to 800 mm and the main rainy season of July and August. Here we would like to explore the possibility of
weekly precipitation forecasting by a data-driven deep learning method.

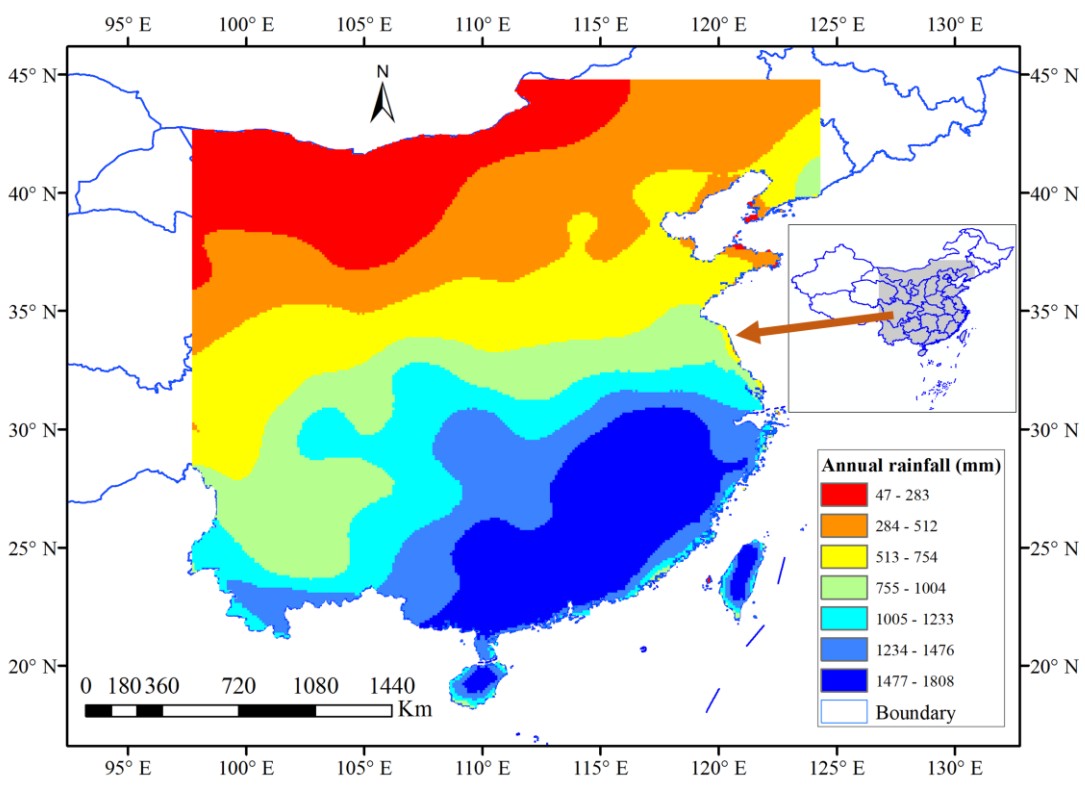


Figure 1: The study area for short-term precipitation forecasting.

Multisource precipitation datasets are used here to obtain the data error and to measure the precipitation forecasting ability of different uncertainty processing strategies, including the Modern-Era Retrospective Analysis for Research and Applications, version 2 (MERRA-2) (Gelaro et al., 2017), the National Centers for Environmental Prediction Reanalysis version 2 (NCEP R2) (Saha et al., 2014) and the European Centre for Medium-Range Weather Forecasts Reanalysis version 5 (ERA-5) (Hersbach et al., 2020) datasets from 1980 to 2020. The surface 2-meter temperature and the geopotential height at 500 hPa datasets are collected from the three datasets accordingly as predictors. All these daily datasets are converted to weekly data and are bilinearly interpolated into 0.25° resolution. In the forecasting process, the temperature and geopotential height predictors in the historical three consecutive weeks are used to forecast the precipitation in the target week. For example, the predictors in the first week, the second week and the third week (P1, P2, P3) are used to forecast the total precipitation in the fourth week.

**3 Methods**

**3.1. Estimation of data uncertainty**

The TCH method (Xu et al., 2020b; Premoli and Tavella, 1993) is used to estimate the uncertainty in temperature, geopotential height and precipitation datasets. The collected three datasets are all reanalysis data, which is generated from different physical models and data assimilation algorithms. The different reanalysis datasets and their errors are generally not closely correlated and are regarded as collocated datasets for the uncertainty estimation, similar with existing studies (Xu et al., 2021b; Mccoll et al., 2014; Gruber et al., 2017). In the TCH algorithm, one arbitrary dataset is chosen as the reference among the three datasets, and then the differencing operation is conducted between the reference and the other two datasets to get the differencing series. The covariance of the differencing series is connected to the variance-covariance matrix of precipitation datasets through matrix transformation. The parameters of the variance-covariance matrix are iteratively resolved by minimizing the global correlation of the covariance of the differencing series. A detailed introduction of TCH method could refer to Premoli and Tavella (1993) and Xu et al. (2020b).

Consider the data series $\{P_i, i=1, 2, \ldots, N\}$, they can be generally expressed as

$$P_i = P_{true} + \varepsilon_i \tag{1}$$

where $P_{true}$ denotes the true predictor or predictand data in a specific area and is unknown; $\varepsilon_i$ is the measurement error. Choosing an arbitrary dataset as the reference, the differences of $N$-1 data and the reference data are calculated.

$$D_{iN} = P_i - P_N = \varepsilon_i - \varepsilon_N, \quad i = 1, \ldots, N-1 \tag{2}$$

where $P_N$ is the reference series and the TCH result is independent of the selection of reference data. These $N$-1 differences are

concatenated into an $M \times (N\text{-}1)$ matrix
$$D = [D_{1N} \ D_{2N} \ \cdots \ D_{(N-1)N}] \tag{3}$$
where the rows of $D$ denote the differenced time series with $M$ length. The variance/covariance of $D$ is then obtained by
$$S = \left[\frac{1}{M-1}\right][(D - \bar{D})^T (D - \bar{D})] \tag{4}$$
where $\bar{D}$ represents the average of $D$. The covariance of $D$ can be related to $S$ by
$$S = K^T \cdot R \cdot K, \ \ K = \begin{bmatrix} I \\ -u^T \end{bmatrix} \tag{5}$$
where $R$ is the $N \times N$ covariance matrix of $\{\varepsilon_i, i=1,2,\ldots,N\}$ and $u$ represents the vector $[1 \ \cdots \ 1]^T$. The covariance matrix of $D$
is regarded as the Allan covariance when TCH was initially proposed to evaluate the instability of clocks, which requires a
filtering operation on the original time series before differencing. However, the covariance can also be the commonly
mentioned sample variance. The Equation (5) can be reformatted as
$$S = [I - u]\begin{bmatrix} \hat{R} & r \\ r^T & r_{NN} \end{bmatrix}\begin{bmatrix} I \\ -u^T \end{bmatrix} \tag{6}$$
where $\hat{R}$ is the $(N\text{-}1) \times (N\text{-}1)$ matrix and $r$ is the $(N\text{-}1)$ vector $[r_{1N} \ r_{2N} \ \cdots \ r_{N-1,N}]$ grouping covariance estimates with the $N$th
time series and $r_{NN}$ denotes the variance of $N$th time series. This partitioning can help solve the underdetermined problem in
Equation (5) by isolating the $N$ free parameters ($r$ and $r_{NN}$). When the free parameters are determined, the unknown elements
in $\hat{R}$ is obtained by
$$\hat{R} = S - r_{NN}[uu^T] + ur^T + ru^T \tag{7}$$
A suitable objective function is suggested by Galindo and Palacio (1999) based on the Kuhn-Tucker theorem to minimize
the quadratic mean of covariances
$$F(r, r_{NN}) = \sum_{i<j} \frac{r_{ij}^2}{L^2} \tag{8}$$
subjecting to a constraint
$$G(r, r_{NN}) = -\frac{r_{NN} - [r - r_{NN}u]^T \cdot S^{-1} \cdot [r - r_{NN}u]}{L} < 0 \tag{9}$$
where $L = \sqrt[N-1]{\det(S)}$. The initial conditions are selected as follows to meet the constraint (Torcaso et al., 1998).
$$r_{iN}^{(0)} = 0, i < N \ \ and \ \ r_{NN}^{(0)} = (2 \cdot u^T \cdot S^{-1} \cdot u)^{-1} \tag{10}$$
Once the free parameters are determined, the unknown elements in $R$ can be solved using Equation (7).
The uncertainties of the predictors and predictands are estimated weekly by the TCH method. The weekly datasets are
grouped according to the weekly climatology and then used to estimate the uncertainty. For example, all the precipitation
datasets which belongs to the first week of each year are concatenated to apply the TCH method in order to get the uncertainty
of the datasets on the first week of each year. Similarly, the data uncertainty on the second week, third week and until the fifty-
two week is evaluated sequentially. This strategy enables a time-variant uncertainty estimation, which is more reasonable as
the precipitation climatology is different for different seasons. The NCEP R2 and ERA-5 data are used to assist the uncertainty
estimation of MERRA-2 data by the TCH method, and the precipitation forecasting experiments are conducted based on
MERRA-2 data to evaluate the proposed forecasting framework.

**3.2. Variational Bayesian inference**

Here we introduce the variational inference theory (Hoffman et al., 2013), which is a standard Bayesian modeling
technique for the estimation of model uncertainty. Given the input data $X=\{x_1,...,x_N\}$ and the output data $Y=\{y_1,...,y_N\}$, the
Bayesian regression is to find suitable parameters within the function $y=f^w(x)$ which could generate the output $Y$ according to
the input $X$. The parameters $w$ is assumed to obey a prior distribution $p(w)$ before the observations are known. When the
observed data is obtained, it is possible to determine which parameters are more suitable for the function according to the data.
A likelihood distribution $p(y|x,w)$ is defined to describe the probability of $y$ generated by $x$ and $w$. For example, a Gaussian
likelihood function is defined as
$$p(y|x,w) = \mathcal{N}(y; f^w(x), \tau^{-1}I) \tag{11}$$
where $\tau^{-1}$ is the observation noise and $I$ is the identity matrix.
Given the input data $X$ and the output data $Y$, the Bayesian theorem is to find the posterior distribution of parameters in
the parameter space.
$$p(w|X,Y) = \frac{p(Y|x,w)p(w)}{p(Y|X)} \tag{12}$$
where the numerator $p(Y|X)$ is the normalization factor, also named as model evidence.
$$p(Y|X) = \int p(Y|X,w)p(w)dw \tag{13}$$
The solution of Equation (13) needs to marginalize the likelihood over $w$, which is tractable analytically for some simple
models such as Bayesian regression, while is intractable for complex models such as deep learning methods (Gal, 2016).
Given the new input data $x'$, the forecasted value is generated by the integral of probability over the parameter space,
which is called the inference process.
$$p(y'|x',X,Y) = \int p(y'|x',w)p(w|X,Y)dw \tag{14}$$
Since the posterior distribution of parameters $p(w|X,Y)$ cannot be obtained analytically, an approximate analytical
distribution $q_\theta(w)$ could be defined, with $\theta$ as the parameter to be estimated, to be as close as the posterior distribution. The
Kullback-Leibler (K-L) divergence (Kullback and Leibler, 1951) is an indicator to measure the similarity of two distributions,

also known as relative entropy. The objective function is to minimize the K-L divergence between the two distributions.

$$KL(q_\theta(w)||p(w|X,Y)) = \int q_\theta(w) log \frac{q_\theta(w)}{p(w|X,Y)} dw \qquad (15)$$

The optimal variational distribution $q'_\theta(w)$ is obtained when the K-L divergence is minimized. The estimated variational distribution could be regarded as the posterior distribution of parameters and then the predictive distribution could be generated.

$$p(y'|x',X,Y) \approx \int p(y'|x',w)q'_\theta(w)dw =: q'_\theta(y'|x') \qquad (16)$$

The above inference process is the variational Bayesian inference. Variational inference replaces the integral of the likelihood with optimization, which simplifies the estimation of posterior distribution.

**3.3. Monte Carlo sampling**

Monte Carlo method is a kind of stochastic simulation technology, proposed by Stanislaw Ulam and John von Neumann during the second world war (Von Neumann and Ulam, 1951). Monte Carlo methods are used to estimate unknown parameters by random sampling and are widely applied in mathematics, physics, game theory and finance (Brooks, 1998; Jacoboni and Lugli, 2012; Metropolis and Ulam, 1949; Rubinstein and Kroese, 2016).

In Equation (14), the posterior distribution $p(w|X,Y)$ cannot be solved analytically. Assume $U_i$ as the weight matrix $K_i \times K_{i-1}$ from $i$-1 layer to $i$ layer, i.e. $w=\{U_i\}_{i=1,...,L}$, a variational weight distribution $q(w)$ is defined to randomly replace the columns with zero (dropout process).

$$U_i = H_i \cdot diag\left(\left[z_{i,j}\right]_{j=1}^{K_i}\right) \qquad (17)$$

$$\left[z_{i,j}\right] \sim Bernoulli(p_i), i = 1, ..., L, j = 1, ..., K_{i-1} \qquad (18)$$

where $p_i$ and $H_i$ are variational parameters; $z_{i,j}$ is a binary variable, with a value of zero representing the abandoning of $j$th unit in $i$-1 layer and a value of one the keeping, based on the *Bernoulli* distribution at the probability $p_i$.

The predictive distribution is estimated after minimizing the K-L divergence.

$$q(y'|x') = \int p(y'|x',w)q(w)dw \qquad (19)$$

The predictive mean and variance can be obtained after repeating the dropout process multiple times.

$$\mathbb{E}_{q(y'|x')}(y') = \int y' q(y'|x')dy' = \int y' \mathcal{N}(y'; \hat{y}'(x', U_1, ..., U_L), \tau^{-1}I)Bern(U_1)\cdots Bern(U_L)dU_1 \cdots dU_L dy' =$$

$\quad \int \hat{y}'(x', U_1, \ldots, U_L) Bern(U_1) \cdots Bern(U_L) dU_1 \cdots dU_L \approx \frac{1}{T} \sum_{t=1}^{T} \hat{y}'(x', \hat{U}_{1,t}, \ldots, \hat{U}_{L,t})$ (20)
$\quad Var_{q(y'|x')}(y') \approx \tau^{-1}I + \frac{1}{T}\sum_{t=1}^{T} \hat{y}'(x', \hat{U}_{1,t}, \ldots, \hat{U}_{L,t})^T \hat{y}'(x', \hat{U}_{1,t}, \ldots, \hat{U}_{L,t}) - \mathbb{E}_{q(y'|x')}(y')^T \mathbb{E}_{q(y'|x')}(y')$ (21)
where $u_{t,i}$ is the forecasted value for $i$th pixel and $t$th ensemble. The calculation of predictive variance is based on the standard
deviation of the ensemble, which represents the spread of the forecasted values.
The above Monte Carlo sampling and dropout process is the Monte Carlo dropout technique, which is used to obtain the
model uncertainty here.
**3.4. Joint data and model uncertainties modeling**
Dropout is a Bayesian method to model the model uncertainty in forecasting (Srivastava et al., 2014). However, the data
uncertainty also needs to be considered. Kendall and Gal (2017) regarded the data uncertainty as a trainable parameter and
jointly considered data and model uncertainties. However, the predictand data uncertainty is ignored and the learning of data
uncertainty increases the number of training parameters. Here we propose an integrated modeling framework to fully
incorporate the data and model uncertainties during the training process (Figure 2). First, the data uncertainties of predictors
and predictands are estimated by the TCH method and are assumed as Gaussian distribution.
$\quad \sigma = TCH(D_i), i = 1,2,3$ (22)
$\quad \sigma_x \sim \mathcal{N}(0, \sigma)$ (23)
$\quad \sigma_y \sim \mathcal{N}(0, \sigma)$ (24)
The sampling methods are used to sample from the data distribution to produce ensemble forecasts. The sampling process
is conducted both for predictor data and predictand data. The data uncertainty is randomly sampled $T$ times to generate an
ensemble of predictors and predictands. In the meantime, the parameters are randomly dropped out for $T$ times to construct a
parametric ensemble. The perturbed data and parameter values are jointly used to calculate the training loss. The objective
function is expressed as follows (Kendall and Gal, 2017), which is obtained from the likelihood of a Gaussian process
(Srivastava et al., 2014).
$\quad \mathcal{L}(\theta, p) = -\frac{1}{N}\sum_{i=1}^{N} \log p\left(y_{i,\sigma} \Big| f^{\hat{U}_i}(x_{i,\sigma})\right) + \frac{1-p}{2N}\|\theta\|^2$ (25)
$\quad \sigma = \sqrt{(\sigma_x^{(l)})^2 + \sigma_y^2}$ (26)
where $N$ is the sample size; $p$ is the dropout probability; $\widehat{U}_i \sim q'_\theta(U)$; $\theta$ is the parameter to be estimated; $\sigma_x$ and $\sigma_y$ are the data
uncertainty for predictor and predictand, respectively. $\sigma_x^l$ represents the propagated data uncertainty for the $l$th network layer.
The negative log-likelihood function can be deduced according to the objective function.
$\quad -\log p\left(y_{i,\sigma}\Big|f^{\widehat{U}_i}(x_{i,\sigma})\right) \propto \frac{1}{2\sigma^2}\left\|y_i - f^{\widehat{U}_i}(x_i)\right\|^2 + \frac{1}{2}\log(\sigma^2)$ $\hfill$ (27)
where $\sigma$ is the regression noise, with the mean of zero in a Gaussian distribution.
$\quad$ The objective function consists of a mean square error (MSE) term adjusted by data uncertainty and a regularization term,
which is the negative logarithm of the Gaussian likelihood function. The objective function includes an uncertainty parameter
$\sigma^2$, which is determined by the sum of propagated uncertainty and target data uncertainty. The minimization of the negative
log-likelihood function could be reached by differentiating the optimization function and setting to zero.
$\quad \frac{\partial}{\partial\sigma^2}\left[\frac{1}{2\sigma^2}\left\|y_i - f^{\widehat{U}_i}(x_i)\right\|^2 + \frac{1}{2}\log(\sigma^2)\right] = 0$ $\hfill$ (28)
$\quad \Longrightarrow -\frac{1}{2\sigma^4}\left\|y_i - f^{\widehat{U}_i}(x_i)\right\|^2 + \frac{1}{2\sigma^2} = 0$ $\hfill$ (29)
$\quad \Longrightarrow \sigma^2 = \left\|y_i - f^{\widehat{U}_i}(x_i)\right\|^2$ $\hfill$ (30)
where the minimum value of the negative log-likelihood function could be reached when the data variance equals to the square
of the difference between the forecasted value and the observation.
$\quad$ Once the network weights are determined according to the objective function, the new input data uncertainty is propagated
and the weights are randomly sampled to produce the forecasted ensemble. The predictive mean and variance are calculated
from the predictive ensembles.
$\quad \mu_i = \frac{1}{T}\sum_{t=1}^{T} y_{t,i}$ $\hfill$ (31)
$\quad Var_i \approx \frac{1}{T}\sum_{t=1}^{T} y_{t,i}^2 - \left(\frac{1}{T}\sum_{t=1}^{T} y_{t,i}\right)^2 + \frac{1}{T}\sum_{t=1}^{T} \sigma_{t,i}^2$ $\hfill$ (32)

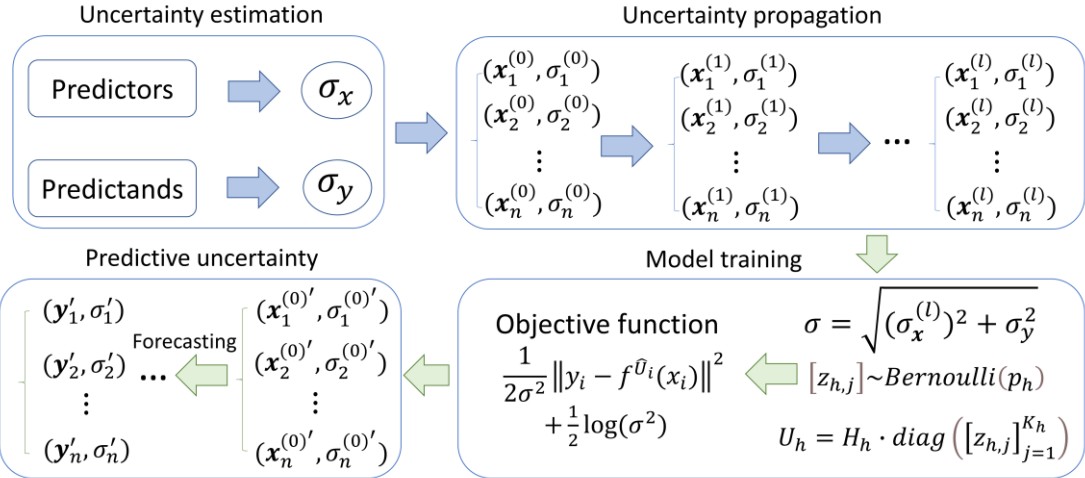


Figure 2: The proposed integrated data-model uncertainty modeling framework in precipitation forecasting. $\sigma_x$ and $\sigma_y$ are the
predictor and predictand uncertainty, respectively. $x_n^l$ and $\sigma_n^l$ represents the propagated data value and uncertainty,
respectively, for the $l$th network layer.

**3.5. The deep learning forecasting framework**

In deep learning, encoder-decoder is a commonly used forecasting model (Badrinarayanan et al., 2017; Cho et al., 2014).
In the encoder process, an input signal is converted into a one-dimension vector with fixed length. In the decoder process, the
one-dimension vector is transformed into the target data with variable length. The available networks used for encoder and
decoder processes are arbitrary and depend on the specific problem, such as convolutional neural network (CNN), recurrent
neural network (RNN) and long-short term memory (LSTM) network (Hochreiter and Schmidhuber, 1997; Goodfellow et al.,
2016). The CNN network is used in this study to construct the encoder-decoder forecasting model. Here we designed a deep
learning encoder-decoder model for weekly precipitation forecasting (Figure 3). The temperature and geopotential height data
for previous three weeks are regarded as inputs, with an image size of 64×64×7 (including the land-sea mask). In the encoder
process, the input image is down-sampled by a series of convolution, pooling and dropout operations, resulting in a one-
dimension vector (1×1×2048). In the decoder process, the one-dimension vector is up-sampled by deconvolution, dropout and
convolution operations, resulting in a forecasted precipitation image (64×64×1). The down-sampling and up-sampling
procedures are used to learn the nonlinear mapping relationships between predictors and predictands.
In the training process, the optimization algorithm is set to Adam (Kingma and Ba, 2014), which is a stochastic learning
algorithm based on adaptive moment. The network learning rate is set to 0.001 and the stopping rule of iteration is that the
validating error does not decrease for at least 100 times. The data uncertainty is propagated forward according to the law of
uncertainty propagation and the dropout process is repeated 10 times with a dropout rate of 0.5. The random seed is set to 1 to
enable the reproducibility of the experiment. The experimental data spans from 1980 to 2020 (2139 weeks), of which 60%,
20% and 20% of the data are used for training, validating and testing, respectively. The optimal model parameters are
determined based on the minimal validating loss.

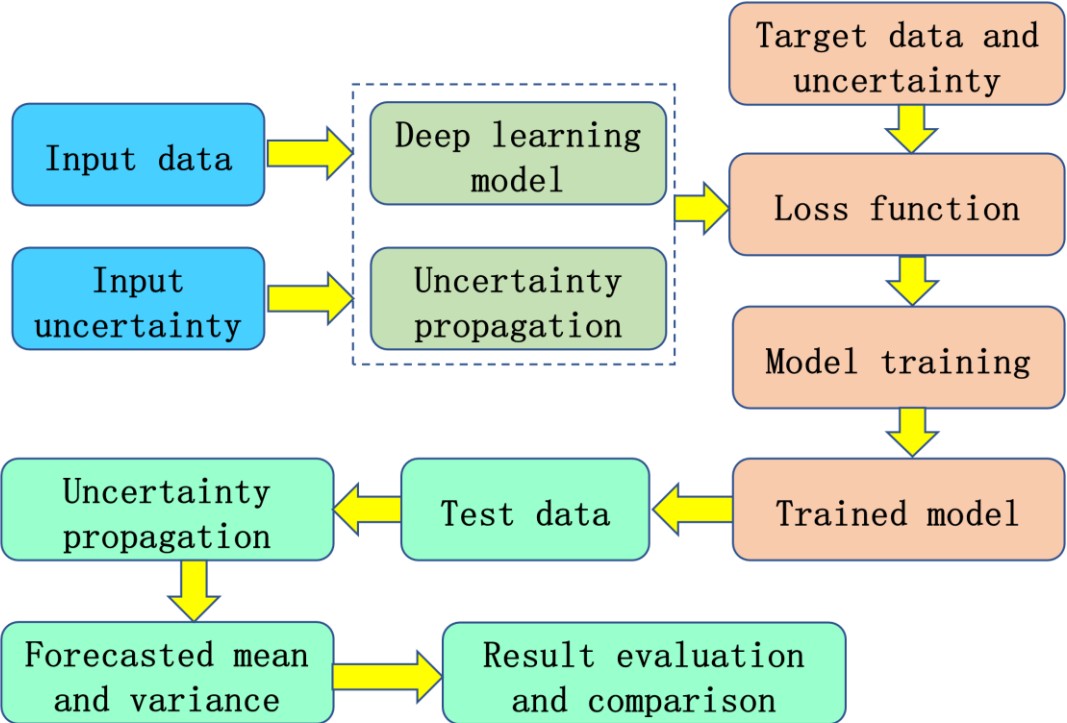


Figure 3: The developed deep learning model for precipitation forecasting.
The proposed deep learning framework for precipitation forecasting is demonstrated in Figure 4. The input data and its
uncertainty are prepared and are considered as inputs for the forecasting model. The model weights are initialized and the input
uncertainty is propagated forward according to the weights. The loss function value is calculated according to the forecasted
value, the propagated uncertainty, target data and its uncertainty. The forecasting model is trained according to the optimization
algorithm and then the trained model is obtained. Next the test data is used to produce the forecasted value and variance based
on the model weights and uncertainty propagation. Finally, the forecasted value and variance are evaluated and compared with
several precipitation forecasting methods.

Figure 4: The proposed deep learning framework for precipitation forecasting.
We designed a series of comparison experiments to investigate the effect of different uncertainty processing strategies on
the forecasting performance (Table 1). The precipitation forecasting experiment without considering uncertainty is used as the
baseline (Experiment 1). The mean square error is used as the loss function and the data and model uncertainties are not
considered in Experiment 1. The uncertainty sources are incorporated differently into the experiments, including predictor
uncertainty (Experiment 2), predictor and predictand uncertainties (Experiment 3), model uncertainty based on Srivastava et

al. (2014)'s method (Experiment 4), data and model uncertainties based on Loquercio et al. 2020)'s method (Experiment 5), and data and model uncertainties (Experiment 6) based on the proposed framework here. The data uncertainty only includes the propagated uncertainty from the input data in Equation (26) in Experiment 2, while the propagated uncertainty from the input data and the target data uncertainty are both included in Experiment 3. In Experiment 4, the data uncertainty is ignored and the model parameters are randomly sampled for 10 times to get the model spread. In Experiment 5, the input data uncertainty is propagated and the model uncertainty is modeled by sampling the parameters. In Experiment 6, the input uncertainty is propagated and the target data uncertainty is included in Equation (26) and the model uncertainty is represented by multiple sampling process.

Table 1. A summary of the experiment setup in this study. The √ and × symbol indicates that the specific factor is considered and omitted, respectively.

| Experiments | Uncertainty processing | | Model uncertainty |
|---|---|---|---|
| | Data uncertainty | | |
| | Input data (predictor) uncertainty | Target data (predictand) uncertainty | |
| Experiment 1 | × | × | × |
| Experiment 2 | √ | × | × |
| Experiment 3 | √ | √ | |
| Experiment 4 (Srivastava et al., 2014) | × | × | √ |
| Experiment 5 (Loquercio et al., 2020) | √ | × | √ |
| Experiment 6 (This study) | √ | √ | √ |

The root mean square error (RMSE) statistic is used to measure the difference between forecasted value and true value.

$$RMSE = \sqrt{\frac{\sum_{i=1}^{n}(y_i - \hat{y}_i)^2}{n}} \tag{33}$$

where $y_i$ is the true value or observation and $\hat{y}_i$ is the forecasted value; $n$ is the sample size.

## 4 Results and discussion

### 4.1. The uncertainty of input and output datasets

The data uncertainty of predictors and predictand is calculated based on the TCH method and is shown in Figure 5. The precipitation data uncertainty is much higher than the temperature and geopotential height variables, with a median of ~43% relative uncertainty fraction for precipitation, ~2% for temperature and ~1% for geopotential height in the MERRA-2 data, and this pattern is also seen in the ERA-5 and NCEP R2 datasets. Therefore, the precipitation data suffer from greater uncertainty relative to the input data and the predictand uncertainty should not be ignored in the training process. The combination of the propagated input uncertainty and the predictand uncertainty is used as the adjusted parameter to regularize the loss function, which is relatively reasonable as the data with larger uncertainty should contribute less to the total training loss. It should be noted that the predictor and predictand data are normalized to [0,1] before uncertainty estimation to ensure a fair comparison of uncertainty value. The high uncertainty for precipitation data is related to strong spatiotemporal heterogeneity of precipitation and the high inconsistency among the reanalysis data (Xu et al., 2020b), while temperature and geopotential height data are much more homogeneous in space and time.

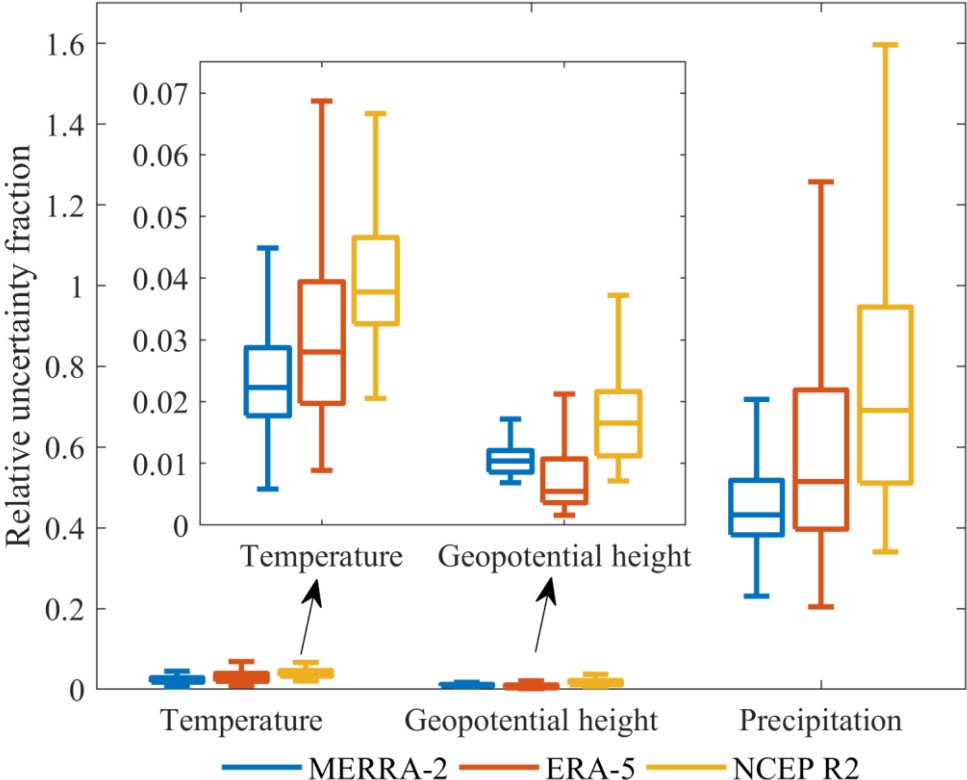

Figure 5: The data uncertainty calculated by the TCH method. The uncertainty distribution is plotted according to the uncertainty over all the pixels of the study area.

**4.2. Overall precipitation forecasting performance**

The RMSE and uncertainty in Table 2 are calculated based on the averaged values of all the grid cells. As for the predictive uncertainty, the forecasting method that only considers model uncertainty (Srivastava et al., 2014) obtains the minimum predictive uncertainty (Table 2). However, the data uncertainty is not considered when only sampling from the parameters and thus the impact of data error on forecasting is not evaluated. Loquercio et al. (2020) used the law of uncertainty propagation to propagate the data uncertainty and sampled the parameters randomly during training. In our proposed method, the input data uncertainty, target data uncertainty and model uncertainty are jointly coupled by uncertainty propagation and random parameter sampling. The average predictive uncertainty (10.859) based on the proposed method is smaller than the Loquercio et al. (2020)'s method (15.232), and the predictive $R^2$ is higher (0.539) than the latter (0.523). In this regard, the proposed method could reduce the predictive uncertainty of precipitation forecasting to some extent, when jointly modeling data and model uncertainties. The proposed method could improve the precipitation forecasting performance and could improve the reliability of precipitation forecasting by reducing the uncertainty.

When only the input uncertainty is modeled in the forecasting model, the predictive uncertainty is 14.729. If the target data uncertainty is coupled with input uncertainty, the predictive uncertainty is substantially reduced (9.290). In Equation (30), when the predictive error on the right side of the equation reaches local minimum and remains unchanged basically, the left side of the equation includes the input uncertainty propagation and the target data uncertainty. When new data is used to make prediction, the predictive uncertainty is generated by the input uncertainty and the law of error propagation. Thus, when only the input uncertainty is modeled in Equation (30), the left side of this Equation equals to the propagated uncertainty from the input data. If the left side of the Equation (30) is replaced from the propagated input uncertainty with the combination of propagated input uncertainty and target uncertainty, the propagated input uncertainty after replacement will be smaller than that of no replacement, i.e. $(\sigma_x^{(l)})^2 = \sigma^2 - \sigma_y^2 < \sigma^2 = \left\| y_i - f^{\hat{\theta}_i}(x_i) \right\|^2$.

The predictive accuracy generally increases with the uncertainty processing procedures. The predictive RMSE decreases when incorporating the predictor uncertainty processing relative to the prediction without uncertainty handling. Further improvement is expected when considering the predictor and predictand uncertainties from the data aspect, relative to the sole predictor treatment. The R-square exhibits slight improvement when incorporating model uncertainty (Srivastava et al., 2014) relative to the prediction without uncertainty processing. In Loquercio et al. (2020)'s method, the input data uncertainty and model uncertainty are both considered and the predictive performance increases relative to the no uncertainty and predictor uncertainty processings. In our method, the input data uncertainty, target data uncertainty and model uncertainty are jointly considered, reaching the lowest RMSE and the highest R-square relative to the above uncertainty processing methods.

Table 2. The accuracy of precipitation forecasting based on different uncertainty processing strategies. The best RMSE and the highest R-square are shown in bold for each column.

| Uncertainty processing | RMSE | Uncertainty | R-square |
|---|---|---|---|
| No uncertainty | 25.138 | - | 0.503 |
| Predictor uncertainty | 25.065 | 14.729 | 0.517 |
| Predictor and predictand uncertainties | 24.679 | 9.290 | 0.533 |
| Model uncertainty (Srivastava et al., 2014) | 25.156 | 1.570 | 0.506 |
| Data and model uncertainties (Loquercio et al., 2020) | 25.056 | 15.232 | 0.523 |
| Data and model uncertainties (This study) | **24.531** | 10.859 | **0.539** |

## 4.3. Spatial patterns of precipitation forecasting

In Figure 6, the spatial patterns of the RMSE for precipitation forecasting demonstrate some similarities and differences between different uncertainty processing strategies. Overall, the spatial distribution of RMSE is similar with each other and is smaller in the northwest region but larger in the southeast region. In the places where the annual rainfall is abundant, the water cycle process is accelerated and the precipitation observations may suffer from large uncertainty. The difficulty of forecasting extreme high precipitation volume also increases the average RMSE in the southeast region relative to the northwest region (Yuan et al., 2012; Huang et al., 2013). There are some differences of the forecasting error among different forecasting methods in local areas. For example, the forecasting performance based on our proposed method could outperform the methods in Experiments 1, 2, 4 and 5 and is comparable with the methods in Experiments 3 for the local areas covered by black circles in Figure 6.

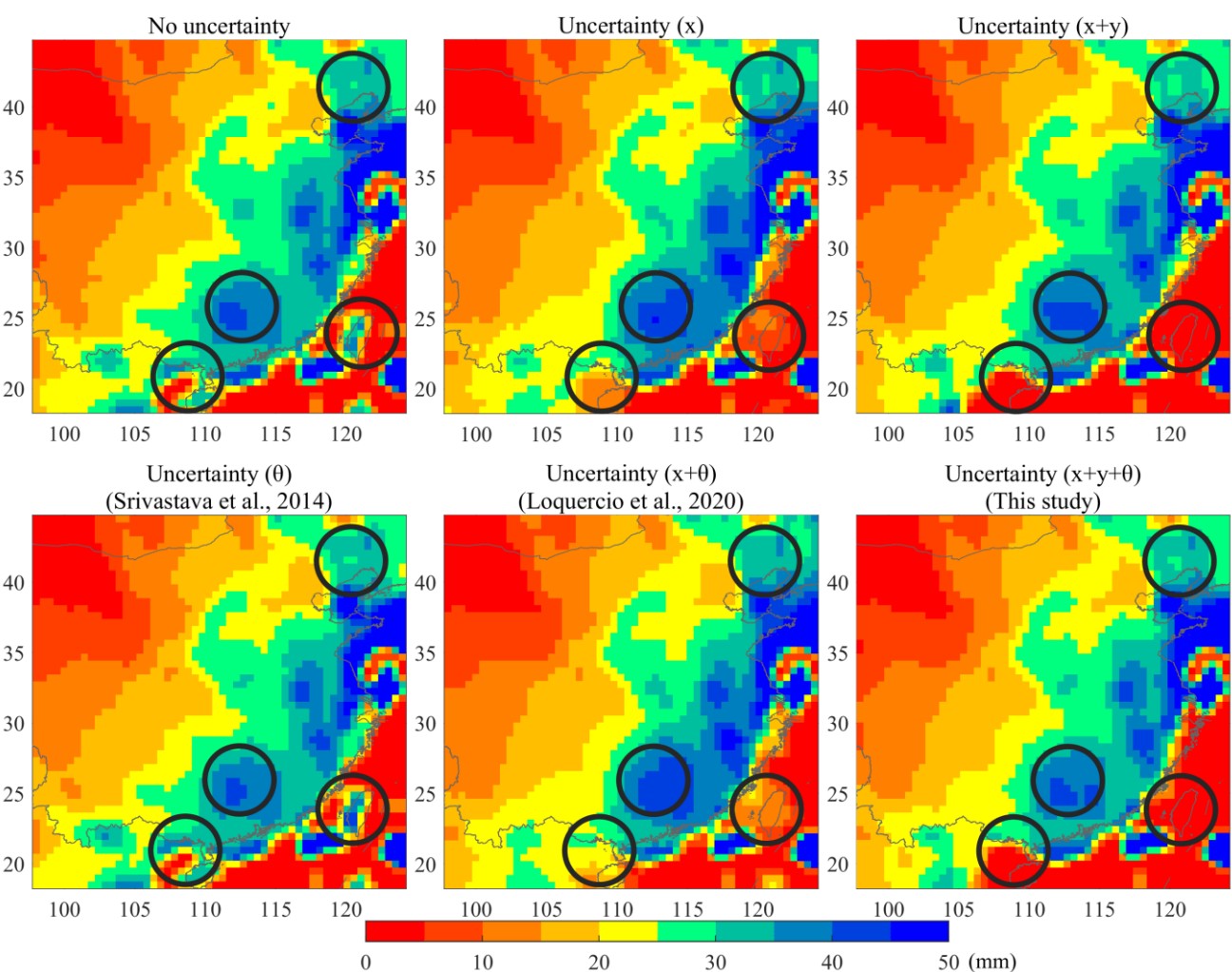

Figure 6: The spatial patterns of RMSE for precipitation forecasting. In this figure, x means the modeling of input uncertainty; x+y represents the modeling of input and output uncertainty; x+y+θ indicates the modeling of input uncertainty, output uncertainty and model uncertainty. The black circles represent the highlighted areas.

Figure 7 demonstrates the difference of the predictive RMSE between other uncertainty processing approaches and the proposed method for precipitation forecasting. It can be seen that there is little difference of the predictive RMSE between the

proposed method and other uncertainty processing approaches. However, the RMSE is larger for some uncertainty processing

methods relative to the proposed method in some areas, including the Hainan and Taiwan provinces and part of eastern China.

It is likely that the predictive RMSE is reduced in the proposed method by improving the prediction performance in the areas

with accelerated water cycle and abundant rainfall. The target data uncertainty is incorporated into the objective function to

pay different attentions in space during the training process, leading to a different weight distribution of the forecasting model

for the proposed method relative to others. The changing weight distribution improves the forecasting accuracy in the southeast

areas by reducing the forecasting error of extreme precipitation.

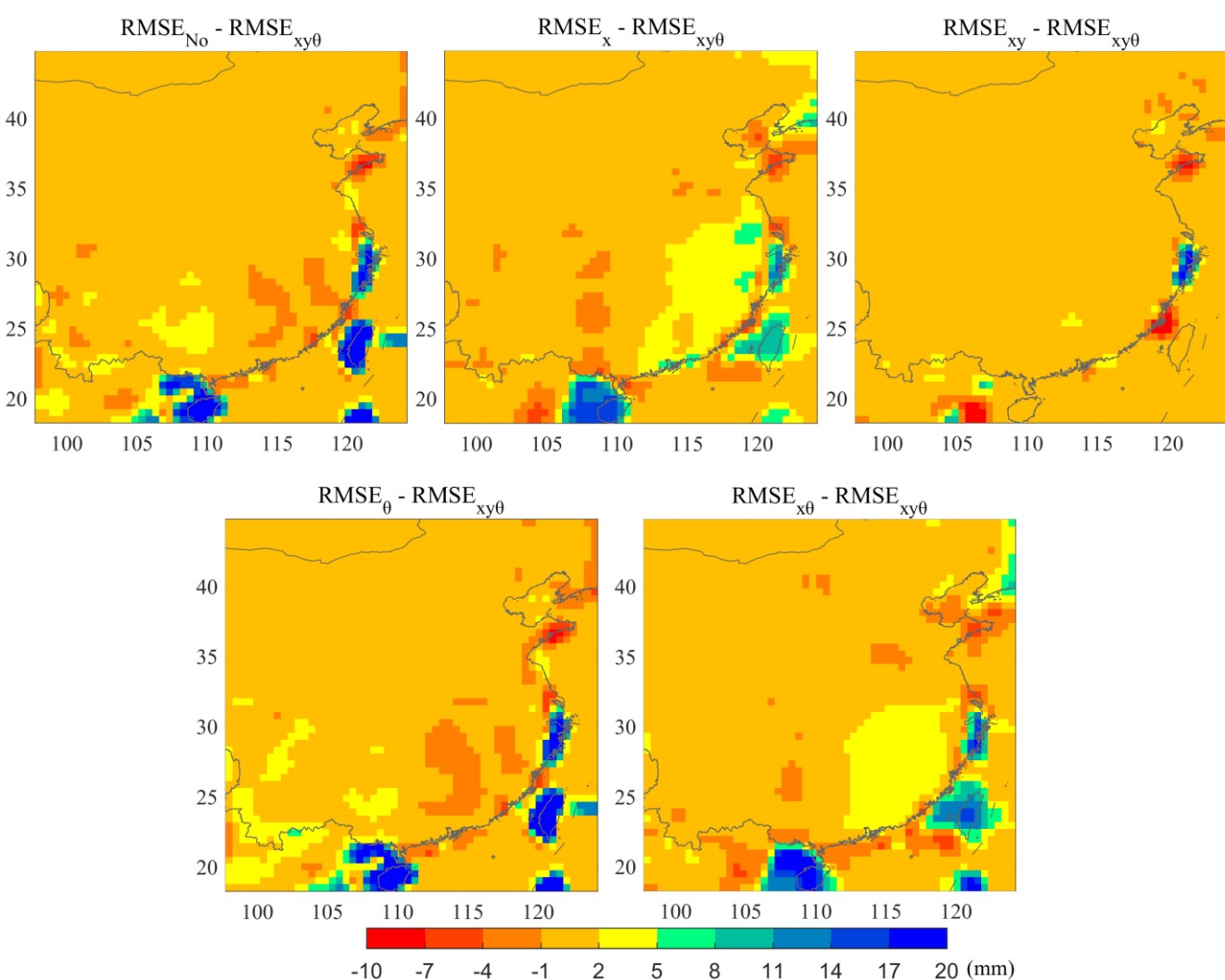

Figure 7: The difference of the predictive RMSE between the proposed method and other uncertainty processing approaches

for precipitation forecasting.

Figure 8 demonstrates the impact of different uncertainty processing methods on the predictive uncertainty of

precipitation. The spatial patterns of predictive uncertainty indicate larger uncertainty in the southern China relative to northern

China, which is consistent with the knowledge that the water cycle in southern China is accelerated and precipitation

forecasting may suffer from large uncertainty. If only the input uncertainty is considered in the forecasting model, the predictive

uncertainty is large in the central and southeast regions. The predictive uncertainty could be substantially reduced when

393 incorporating the target data uncertainty besides the input uncertainty. In Loquercio et al. (2020)'s method, the predictive

394 uncertainty is slightly higher than that of input uncertainty modeling in space, because the input and model uncertainties are

395 jointly modeled. Our proposed method could include the input, target and model uncertainties jointly and could help reduce

396 the predictive uncertainty to a large extent, relative to the Loquercio et al. (2020)'s method.

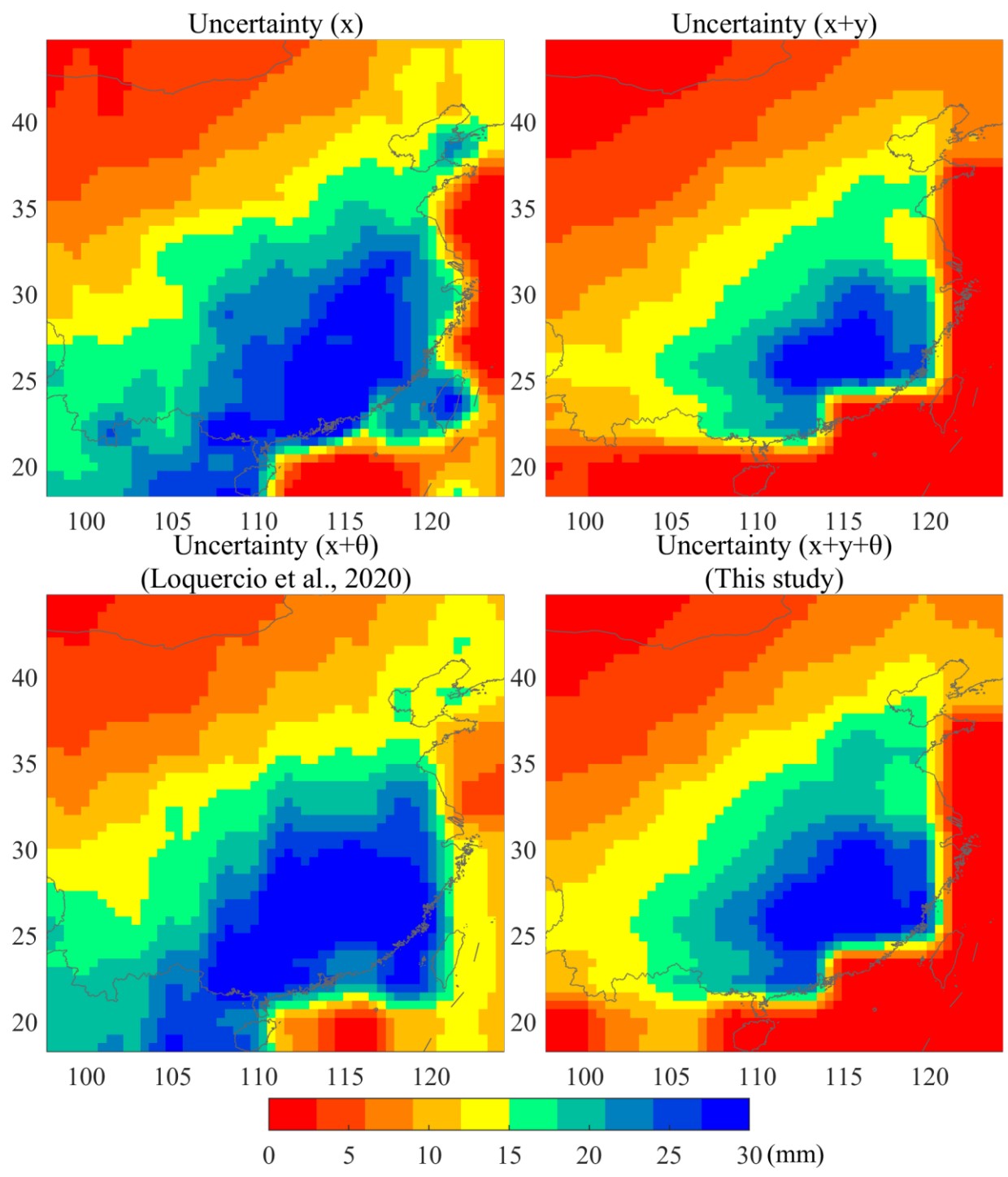

397

398 Figure 8: The spatial patterns of uncertainty for precipitation forecasting.

### 4.4. Uncertainty analysis and discussion

400 In precipitation forecasting, data and model uncertainties both bring uncertainty to the forecasting result. The higher the

data and model uncertainties, the more divergent and less reliable the forecasting. Therefore, the data and model uncertainties should be jointly considered in the forecasting process (Gal, 2016; Kendall and Gal, 2017; Loquercio et al., 2020; Parrish et al., 2012). Although the predictive error is close to each other among different forecasting methods in Figure 6 and Table 2, the predictive uncertainty has some discrepancies. The modeling of input uncertainty only in the forecasting model would bring high predictive uncertainty and the target data uncertainty is ignored. The joint modeling of input and target uncertainties could reduce the predictive uncertainty substantially, which is related to the change of the variance in Equations (26) and (30) corresponding to the minimum value of the forecasting error term. The propagation of input uncertainty is constrained by refining the uncertainty representation in Equation (26) after incorporating the target uncertainty term and thus changing the weight training process.

Figure 9 demonstrates the relationship between the predictive uncertainty by the proposed method and the summation of the predictive uncertainty from data and model. The predictive uncertainty by the proposed method ($\sigma_{xy\theta}$) agrees well with the summation of the predictive uncertainty from data and model, with a regressed slope of 1.004 and a $R^2$ of 0.98, indicating the good consistency of the predictive uncertainty between joint and separate uncertainty modeling. The median predictive uncertainty is 8.04, 1.40 and 9.49 mm for data, model and joint data-model uncertainty modeling, respectively in the precipitation forecasting experiments (Figure 10). The data and model uncertainty account for ~85% and ~15% of the total predictive uncertainty by the proposed method respectively. The model structure is fixed in this study and thus the model uncertainty comes from mainly the parameters. The dropout method enables the random abandoning of network parameters over the specific layer and prevents the overfitting of the forecasting model. The distributions of model weights are changed when incorporating the data uncertainty into the objective function (Figure 11). The model weights become larger and more dispersive relative to the prediction without uncertainty processing or with model uncertainty consideration. The data uncertainty is propagated into the prediction through model weights and thus the data uncertainty contributes more to the predictive uncertainty than model uncertainty. Although the model weights appear more scattered, the predictive accuracy exhibits small improvement due to the uncertainty consideration in the objective function.

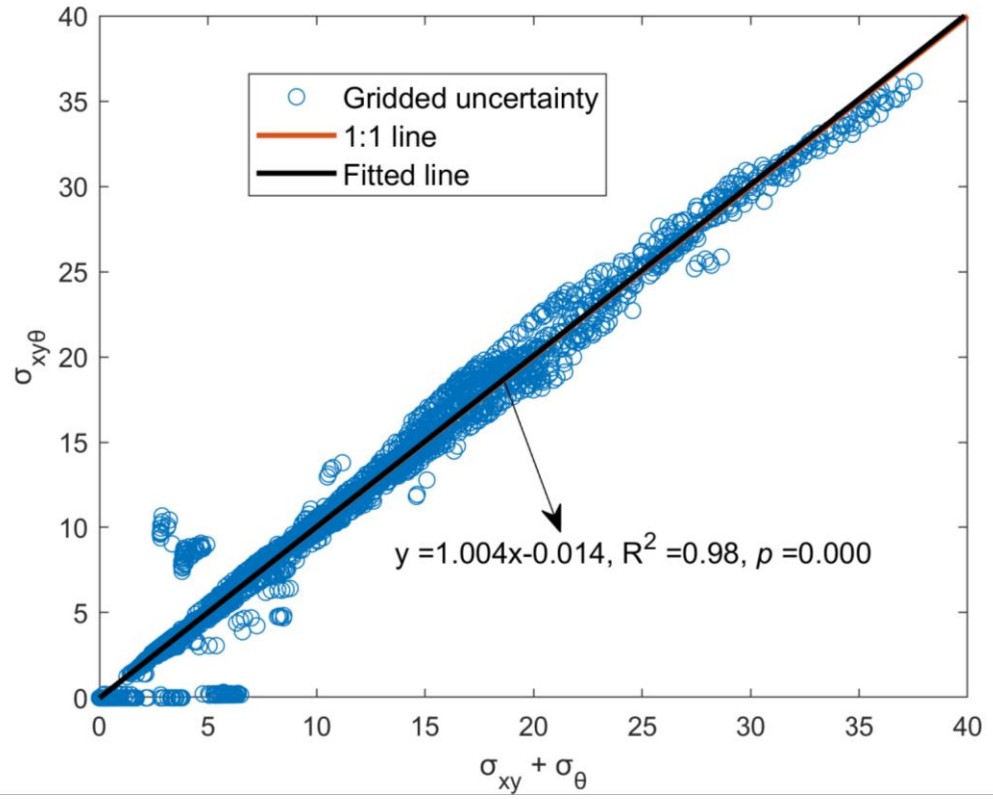

Figure 9: The relationship between the summation of predictive uncertainty from data (input and target data) and model and the predictive uncertainty from joint consideration of data (input and target data) and model.

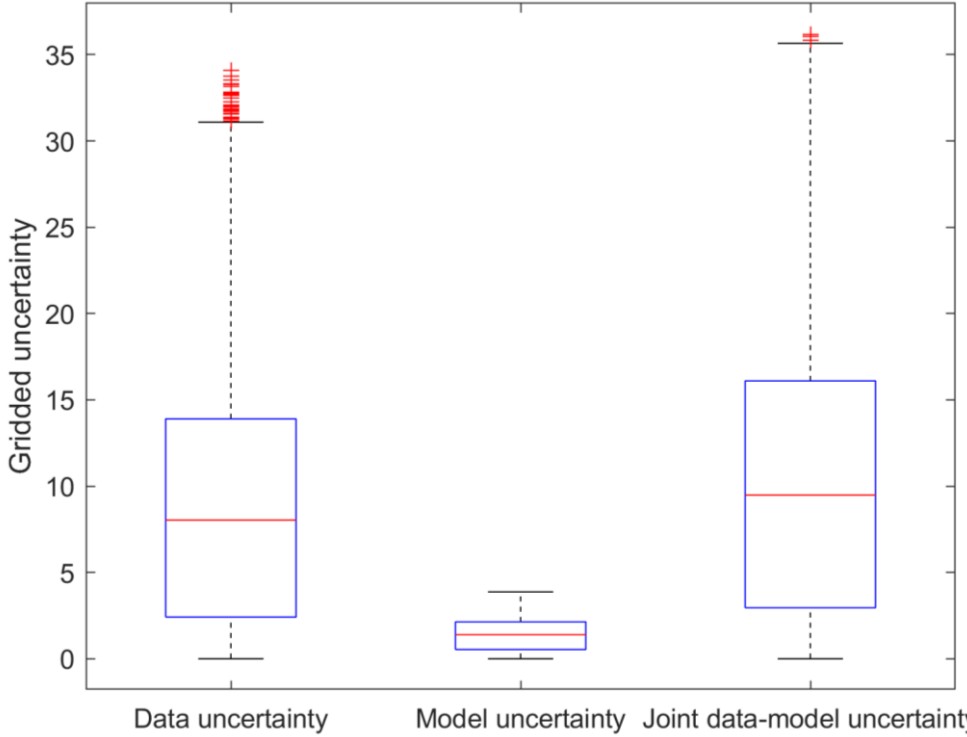

Figure 10: The distributions of the predictive uncertainty from data, model and joint data-model modeling methods for precipitation forecasting.

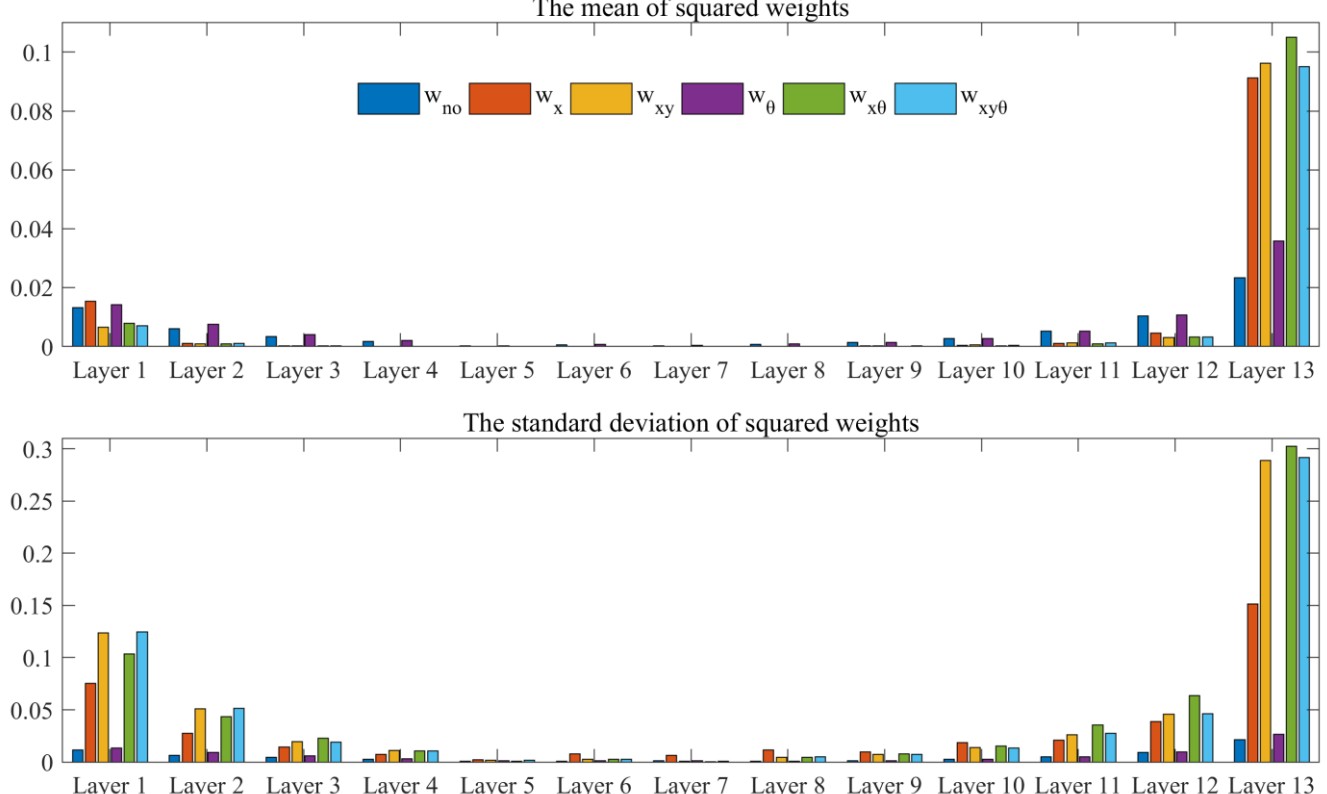

Figure 11: The mean and standard deviation of squared weights of the precipitation forecasting model.

The predictive uncertainty is closely related to the input data uncertainty and model weights. In Figure 2, the input data uncertainty is propagated into predictive uncertainty through model parameters, while the target data uncertainty is incorporated into the training process by the objective function. The incorporation of uncertainty processing changes the training process and the distributions of model weights (Figure 11). The last CNN layer (Layer 13) weights of the precipitation forecasting model are generally higher than the other layers, which may contribute largely to the predictive uncertainty by propagating the data uncertainty. The mean and the standard deviation of the squared weights of the last CNN layer for the Loquercio et al. (2020)'s method are higher than other uncertainty processing methods, suggesting an overall larger and more dispersive weight distribution for Loquercio et al. (2020)'s method than the others, which could partly explain the high uncertainty of the Loquercio et al. (2020)'s method in Table 2.

The proposed method in this study could model the input uncertainty, target uncertainty and model uncertainty jointly and could reduce the predictive uncertainty relative to Loquercio et al. (2020)'s method. The developed method does not increase the training parameter and is a general forecasting uncertainty method for geophysical applications such as temperature forecasting, runoff forecasting and wind speed forecasting, especially for data-driven forecasting models (Ham et al., 2019; Zheng et al., 2020; Hossain et al., 2015).

In numerical precipitation forecasting systems, ensemble forecasting is commonly used to quantify the predictive uncertainty (Duan et al., 2019). In ensemble forecasting, the model parameters and data are perturbed to produce a forecasted ensemble and thus the data and model uncertainties are both considered. However, it would be time-consuming and cost-

expensive to conduct large-sample sampling for complex physical models. In our developed method, the law of error
propagation is used to propagate the data uncertainty. The uncertainty propagation of convolution, max-pooling and
deconvolution in the deep learning forecasting model is tractable in an analytical form. However, the uncertainty propagation
process is generally intractable analytically for complex statistical or physical models. Therefore, the theory and
implementation technology for uncertainty modeling require further development, such as surrogate modeling, Monte Carlo
methods, polynomial chaos expansions and Bayesian approaches (Linde et al., 2017; Sudret et al., 2017; Zhu and Zabaras,
2018; Schiavazzi et al., 2017; Nitzler et al., 2020).
**5 Conclusion**
In this study, we proposed a data-model uncertainty coupling framework to estimate the predictive uncertainty of
precipitation forecasting. In this framework, the predictor and predictand uncertainties are estimated a prior by the TCH method
and are assumed as Gaussian distribution. The predictor uncertainty is propagated forward during training and testing processes
by the law of error propagation. The model uncertainty is represented by randomly abandoning model weights from deep
learning layers. The data and model uncertainties are jointly modeled in the objective function during training and are also
used during the testing process. The loss function is constructed by the MSE statistic adjusted by data uncertainty and a
regularization term based on logarithmic data uncertainty. In the loss function, the adjusting parameter is determined by the
combination of the square of predictor and predictand uncertainties. The forecasted ensembles are used to calculate the
predictive mean and variance to estimate the predictive uncertainty of precipitation.
The weekly precipitation forecasting in southern and northern China is used as an example to examine the effectiveness
of the proposed joint uncertainty modeling framework. Temperature and geopotential height data in previous three weeks are
used to forecast the precipitation in the target week. The forecasting model is developed based on an encoder-decoder CNN
deep neural network, with multivariate spatiotemporal predictor data as inputs and spatiotemporal precipitation data as output.
The experimental results indicate that the proposed joint uncertainty modeling framework for precipitation forecasting exhibits
better forecasting accuracy (improve RMSE by 1-2% and R-square by 1-7% on average) relative to several existing methods,
and could reduce the predictive uncertainty by ~28% relative to Loquercio et al. (2020)'s approach. The reduction of predictive
uncertainty is significative for quantitative precipitation forecasting from a data-driven view. The predictive performance is
improved in the proposed method by incorporating the target data uncertainty and reducing the forecasting error of extreme
precipitation.
The data-driven precipitation forecasting method has limitations in the interpretation part relative to numerical weather
prediction. The precipitation forecasting accuracy for numerical models could still be improved by improving the
parameterization schemes and resolving the uncertainties in observations, parameters and models. The proposed uncertainty

modeling framework may also provide some insights for the uncertainty quantification in numerical prediction models. For example, the uncertainty propagation for the input data and the coupling with target data uncertainty could be used in a data assimilation scheme to estimate the propagated uncertainty in weather forecasting.

Data-driven precipitation forecasting could be used as a tool to assist regional prediction and warning of extreme weather events together with numerical models. The proposed joint data-model uncertainty modeling framework could help estimate the forecasting spread and is a general approach to derive predictive uncertainty for geophysical forecasting applications. Further research should focus on the non-Gaussian uncertainty modeling for complex integrated statistical-physical models.

**Data availability**

The meteorological data are publicly available and can be obtained via the website https://gmao.gsfc.nasa.gov/reanalysis/MERRA-2/ for MERRA-2, https://psl.noaa.gov/data/gridded/data.ncep.reanalysis2.html for NCEP R2 and https://climate.copernicus.eu/climate-reanalysis for ERA-5.

**Author contributions**

LX conceptualized and wrote the paper. CY provided supervision of this study. NC gave support in developing the manuscript. HY and ZC reviewed and revised the manuscript.

**Competing interests**

The authors declare that they have no conflict of interest.

**Acknowledgment**

This research was supported by the National Key Research and Development Program for Young Scientist (2021YFF0704400), the National Key R&D Program (2018YFB2100500), the National Natural Science Foundation of China program (41890822), the National Natural Science Foundation of China (42101429) and the Fundamental Research Funds for the Central Universities, China University of Geosciences (Wuhan) (No. 162301212687).

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
