# Peer review of "Quantifying the uncertainty of precipitation forecasting using"

_Hydrology and Earth System Sciences, 2021_

## Author Comment (AC1)

**Referee comment on "Quantifying the uncertainty of precipitation forecasting using probabilistic deep learning" by Lei Xu et al., Hydrol. Earth Syst. Sci. Discuss., https://doi.org/10.5194/hess-2021-432-RC1, 2021**

**In the presented manuscript, a joint uncertainty modeling method is proposed. The input data uncertainty, target data uncertainty and model uncertainty are jointly modeled in a deep learning precipitation forecasting framework to estimate the predictive uncertainty. The results show that the proposed method can improve precipitiation forecasting accuracy and reduce predictive uncertainty. Having said that I am lost and confused. Here are some of my major concern regarding the presented study:**

**Response:** Thank you very much for the reviewing of our manuscript.

**Precipitation forecasting datasets are usually daily. In this study, the authors use three datasets are all daily. Why the authors convert the data to weekly data? In Line 134, the historical three consecutive weeks are used to forecast the precipitation in the target week. How to determine the "three" weeks?**

**Response:** The weekly precipitation forecasting is used as a case to demonstrate the efficiency of the uncertainty quantification method. The daily data is converted into weekly data because the weekly averaged datasets are used as predictors and the weekly total precipitation is regarded as the predictand in the forecasting experiment based on deep learning method. The historical three consecutive weeks are used to forecast the precipitation in the target week. For example, the predictors in the first week, the second week and the third week ($P_1$, $P_2$, $P_3$) are used to forecast the total precipitation in the fourth week.

**Why the authors use NCEP R2, ERA-5 and MERRA-2 data? For exsample, NCEP CFSv2 also have weekly precipitation forecasting data. In this study, MERRA-2 is used as the reference data. Do different reference data in the uncertainty estimation?**

**Response:** The three datasets, NCEP R2, ERA-5 and MERRA-2, are used to obtain the prior uncertainty estimation of predictors and predictands based on the three-cornered hat method. The uncertainty estimation by the three-cornered hat method is independent on the selection of the reference dataset. Therefore, any of the three datasets can be the reference data for uncertainty estimation.

**How to determine the structure of the deep learning model in Figure 3? Besides the model parameters, the model structure can also generate uncertainty. Have the authors considered this part of uncertainty in this study?**

**Response:** The model structure is constructed based on Convolutional Neural Networks (CNNs). We will plot the detailed model structure in the revised manuscript. We agree that the model structure can also generate uncertainty in forecasting. Since the model structure is constructed

by CNNs, the structural uncertainty comes from to what extent the model structure can represent the precipitation evolution process. In this case, the model structure uncertainty is not considered in this study. We use the dropout techniques in the deep learning model and the neural network cells are randomly dropped from the complete model by a certain probability. The model uncertainty is derived by constructing an ensemble of forecasts by dropout method.

**Lines 423-424: In the places where the annual rainfall is abundant, the water cycle process is accelerated and the precipitation observations may suffer from large uncertainty. Why this uncertainty dosen't exhibit in Figure 7? There are no larger uncertainty observed in the southern China in Figure 7. Is it contradictory?**

**Response:** The precipitation observations in southern China may suffer from larger uncertainty than that of northern China because of the accelerated water cycle for the former region. Figure 6 demonstrates the spatial patterns of the root mean square error (RMSE) for precipitation forecasting. The RMSE is larger for southern China than northern China, which is consistent with the underlying phenomenon that the observational uncertainty is possibly larger for the former than the latter. However, the predictive uncertainty is not necessarily dependent on the predictive RMSE and is not absolutely dependent on the observational uncertainty. The predictive uncertainty is dependent on data uncertainty, model uncertainty, training process, uncertainty modeling and so on. Therefore, the difference of the spatial patterns between predictive RMSE and predictive uncertainty is not contradictory. This difference is also similar with the difference between accuracy and precision. We will discuss the spatial patterns of predictive uncertainty in detail in the revised version.

**How to calculate RMSE and uncertainty in Table 1? Are they the average of all the grid cells? How dose the uncertainty processing to improve forecasting accuracy and reduce predictive uncertainty? Why some methods have considered uncertainty processing but their RMSE increase compared with the no-uncertainty method? Loquercio's method also considers the data and model uncertainties. Why the uncertainty of Loquercio's method is so large?**

**Response:** The RMSE and uncertainty in Table 1 are calculated based on the averaged values of all the grid cells. The proposed method exhibits comparable forecasting accuracy with existing methods as the differences of the RMSE values are very small between the used methods for precipitation forecasting. The predictive uncertainty in this study is smaller than the Kendall and Gal (2017)'s and Loquercio et al. (2020)'s methods, because the objective function is designed by jointly considering input data uncertainty, target data uncertainty and model uncertainty. The uncertainty modeling methods quantify the predictive uncertainty, but may not reduce the RMSE because the RMSE is calculated based on the difference between observations and predictions while the predictive uncertainty is dependent on the predictive spread.

There are two differences between Loquercio's method and our method. One is that the data uncertainty is estimated by the three-cornered hat method, while the data uncertainty is assumed as unknown parameters in Loquercio's method. Another difference is that the input data uncertainty and model uncertainty are considered in the objective function in Loquercio's method, while the target data uncertainty is not considered. However, the target data

uncertainty is included in our method besides the input data uncertainty and model uncertainty. The two differences lead to the different model parameters and data uncertainty estimation. We will discuss these differences in the revised manuscript in detail.

**The section "results" is too brief. The authors may analyse how the proposed framework improve RMSE and uncertainties in detail, for example, the contribution of model uncertainty, input data uncertainty and target data uncertainty.**

**Response:** Thanks for your suggestions. We will expand the results and discussion sections and analyze how the proposed framework influence the RMSE and uncertainties in the revised version. The contribution of model uncertainty, input data uncertainty and target data uncertainty to the predictive uncertainty will also be analyzed.

**Minor comments:**
**The abstract is not complete in my point of view. There are four parts in the abstract usually, i.e., background, method, results and conclusion. The results and conclusion are missing in the abstract.**

**Response:** Thank you for the instructive comments. We will include the results and conclusion parts in the abstract in the revised version.

**Line 150. Is it seasonal or weekly?**

**Response:** Thanks for pointing out this error. It's weekly.

**Line 156. NECP or NCEP?**

**Response:** It should be NCEP.

**Lines 268-269. This sentence is unclear; what is the point the authors want to make with it?**

**Response:** We would like to express that the sampling methods are used to sample from the data distribution to produce ensemble forecasts. The sampling process is conducted both for predictor data and predictand data. We will clarify the expression in the revised version.

**There are many symbols without introduction in Figure 2, such as $x_n^l$, $\sigma_n^l$, etc. The $x_n^l$ is also seen in Equation 16 without introduction.**

**Response:** Sorry for the unclear clarification. The unclarified symbols will be explained in the revised manuscript.

**Lines 319-320. Which deep learning network is used in this study? CNN, RNN, LSTM or all of them? Please make it clear.**

**Response:** The CNN network is used in this study. This will be clarified in the revision.

**Line 376. Typographical error.**

**Response:** This error will be corrected in revision.

**Lines 453-454. Please rephrase.**

**Response:** This sentence is rephrased as "The higher the data and model uncertainties, the more divergent and less reliable the forecasting".

---

## Author Comment (AC2)

**Referee comment on "Quantifying the uncertainty of precipitation forecasting using probabilistic deep learning" by Lei Xu et al., Hydrol. Earth Syst. Sci. Discuss., https://doi.org/10.5194/hess-2021-432-RC2, 2022**

**This study attempts to improve the accuracy of precipitation forecasting by jointly considering multi-source data-model uncertainties in deep learning based modeling framework. A case study conducted in the southern and northern China showed that the developed modeling framework is effective to reduce the uncertainty in precipitation forecasting. In my opinion, this study is valuable and the methodology developed is based on rigorous mathematical formulas that is worthy of recognition. Some of my main comments are listed below**

**Response:** Thank you very much for the reviewing of our manuscript.

**Line 9-26. It is suggested that some summative results should be added to the Abstract.**

**Response:** Thanks for your suggestion. We will include the key results in the abstract in the revision.

**Line 138-159. Adding some key formulas about the TCH algorithm can facilitate the understanding of whole framework.**

**Response:** Thanks. The key formulas for TCH method will be added in the manuscript.

**Some variables in equations need further explanation, such as the 'I' in Equation (1).**

**Response:** We will check all the equations and symbols to make sure that all the variables are explained clearly.

**Line 209. Typographical error. Should be 'estimated'**

**Response:** This error will be corrected.

**Line 349-366. There are many experimental settings. It is suggested to explain them in bullet points, or use a clearer presentation.**

**Response:** Thanks for your suggestion. The bullet points or a summary table will be used to clearly express the experimental settings.

**Line 377. In Figure 5, it is recommended to plot the uncertainty estimation results of all datasets for visual comparison.**

**Response:** Sure. The uncertainty estimation for all datasets will be plotted in the revision.

**Line 402: Add two numbers estimated by Loquercio et al. (2020)'s and Srivastava et al. (2014)'s methods for an intuitive comparison.**

**Response:** Thanks for your suggestion. The specific mathematical values will be added in this sentence for intuitive comparison.

**The Results Section lacks some detailed analysis on how the developed method can improve the prediction accuracy.**

**Response:** We will add detailed analysis on how the proposed framework influences the predictive results, including the RMSE and uncertainty.

---

## Author Response (AR1)

**Referee comment on "Quantifying the uncertainty of precipitation forecasting using probabilistic deep learning" by Lei Xu et al., Hydrol. Earth Syst. Sci. Discuss., https://doi.org/10.5194/hess-2021-432-RC1, 2021**

**In the presented manuscript, a joint uncertainty modeling method is proposed. The input data uncertainty, target data uncertainty and model uncertainty are jointly modeled in a deep learning precipitation forecasting framework to estimate the predictive uncertainty. The results show that the proposed method can improve precipitiation forecasting accuracy and reduce predictive uncertainty. Having said that I am lost and confused. Here are some of my major concern regarding the presented study:**

**Response:** Thank you very much for the reviewing of our manuscript. We carefully address your concerns by re-doing the experiments, enriching and revising the analysis and discussions.

**Precipitation forecasting datasets are usually daily. In this study, the authors use three datasets are all daily. Why the authors convert the data to weekly data? In Line 134, the historical three consecutive weeks are used to forecast the precipitation in the target week. How to determine the "three" weeks?**

**Response:** The weekly precipitation forecasting is used as a case to demonstrate the efficiency of the uncertainty quantification method. The daily data is converted into weekly data because the weekly averaged datasets are used as predictors and the weekly total precipitation is regarded as the predictand in the forecasting experiment based on deep learning method. The predictors in the historical three consecutive weeks are used to forecast the precipitation in the target week. For example, the predictors in the first week, the second week and the third week $(P_1, P_2, P_3)$ are used to forecast the total precipitation in the fourth week. This is clarified in the revised manuscript.

**Why the authors use NCEP R2, ERA-5 and MERRA-2 data? For exsample, NCEP CFSv2 also have weekly precipitation forecasting data. In this study, MERRA-2 is used as the reference data. Do different reference data in the uncertainty estimation?**

**Response:** The three datasets, NCEP R2, ERA-5 and MERRA-2, are used to obtain the prior uncertainty estimation of predictors and predictands based on the three-cornered hat method. The uncertainty estimation by the three-cornered hat method is independent on the selection of the reference dataset. Therefore, any of the three datasets can be the reference data for uncertainty estimation.

**How to determine the structure of the deep learning model in Figure 3? Besides the model parameters, the model structure can also generate uncertainty. Have the authors considered this part of uncertainty in this study?**

**Response:** The model structure is constructed based on Convolutional Neural Networks (CNNs). We plotted the detailed model structure in the revised manuscript (Figure 3). We agree that the model structure can also generate uncertainty in forecasting. Since the model structure is constructed by CNNs, the structural uncertainty comes from to what extent the model structure can represent the precipitation evolution process, which can be represented by the accuracy of forecasting (e.g. the R-square). As the model structure is fixed here, the parametric uncertainty is the major uncertainty for this model. The parametric uncertainty is derived by constructing an ensemble of forecasts by dropout method.

[Figure]

Figure 3: The developed deep learning model for precipitation forecasting.

**Lines 423-424: In the places where the annual rainfall is abundant, the water cycle process is accelerated and the precipitation observations may suffer from large uncertainty. Why this uncertainty dosen't exhibit in Figure 7? There are no larger uncertainty observed in the southern China in Figure 7. Is it contradictory?**

**Response:** We checked the programming code and found a mistake in latitude processing when

presenting the uncertainty. The spatial patterns of uncertainty for precipitation forecasting are shown blow after correcting the error. The spatial uncertainty of precipitation forecasting is listed as follow (Figure 8 in the revised manuscript). The spatial patterns of predictive uncertainty indicate larger uncertainty in the southern China relative to northern China, which is consistent with the knowledge that the water cycle in southern China is accelerated and precipitation forecasting may suffer from large uncertainty. This is clarified in the revised manuscript.

[Figure]

Figure 8: The spatial patterns of uncertainty for precipitation forecasting.

**How to calculate RMSE and uncertainty in Table 1? Are they the average of all the grid cells?**

**How dose the uncertainty processing to improve forecasting accuracy and reduce predictive uncertainty? Why some methods have considered uncertainty processing but their RMSE increase compared with the no-uncertainty method? Loquercio's method also considers the data and model uncertainties. Why the uncertainty of Loquercio's method is so large?**

**Response:** The RMSE and uncertainty in Table 1 are calculated based on the averaged values of all the grid cells. The predictive accuracy generally increases with the uncertainty processing procedures. The predictive RMSE decreases when incorporating the predictor uncertainty processing relative to the prediction without uncertainty handling. Further improvement is expected when considering the predictor and predictand uncertainties from the data aspect, relative to the sole predictor treatment. The R-square exhibits slight improvement when incorporating model uncertainty (Srivastava et al., 2014) relative to the prediction without uncertainty processing. In Loquercio et al. (2020)'s method, the input data uncertainty and model uncertainty are both considered and the predictive performance increases relative to the no uncertainty and predictor uncertainty processings. In our method, the input data uncertainty, target data uncertainty and model uncertainty are jointly considered, reaching the lowest RMSE and the highest R-square relative to the above uncertainty processing methods.

Table 2. The accuracy of precipitation forecasting based on different uncertainty processing strategies. The best RMSE and the highest R-square are shown in bold for each column.

| Uncertainty processing | RMSE | Uncertainty | R-square |
|---|---|---|---|
| No uncertainty | 25.138 | - | 0.503 |
| Predictor uncertainty | 25.065 | 14.729 | 0.517 |
| Predictor and predictand uncertainties | 24.679 | 9.290 | 0.533 |
| Model uncertainty (Srivastava et al., 2014) | 25.156 | 1.570 | 0.506 |
| Data and model uncertainties (Loquercio et al., 2020) | 25.056 | 15.232 | 0.523 |
| Data and model uncertainties (This study) | **24.531** | 10.859 | **0.539** |

[Figure]

Figure 11: The mean and standard deviation of squared weights of the precipitation forecasting model.

The predictive uncertainty is closely related to the input data uncertainty and model weights. In Figure 2, the input data uncertainty is propagated into predictive uncertainty through model parameters, while the target data uncertainty is incorporated into the training process by the objective function. The incorporation of uncertainty processing changes the training process and the distributions of model weights (Figure 11). The last CNN layer (Layer 13) weights of the precipitation forecasting model are generally higher than the other layers, which may contribute largely to the predictive uncertainty by propagating the data uncertainty. The mean and the standard deviation of the squared weights of the last CNN layer for the Loquercio et al. (2020)'s method are higher than other uncertainty processing methods, suggesting an overall larger and more dispersive weight distribution for Loquercio et al. (2020)'s method than the others, which could partly explain the high uncertainty of the Loquercio et al. (2020)'s method in Table 2.

Figure 7 demonstrates the difference of the predictive RMSE between other uncertainty processing approaches and the proposed method for precipitation forecasting. It can be seen that there is little difference of the predictive RMSE between the proposed method and other uncertainty processing approaches. However, the RMSE is larger for some uncertainty processing methods relative to the proposed method in some areas, including the Hainan and Taiwan provinces and part of eastern China. It is likely that the predictive RMSE is reduced in the proposed method by improving the prediction

performance in the areas with accelerated water cycle and abundant rainfall. The target data uncertainty is incorporated into the objective function to pay different attentions in space during the training process, leading to a different weight distribution of the forecasting model for the proposed method relative to others. The changing weight distribution improves the forecasting accuracy in the southeast areas by reducing the forecasting error of extreme precipitation.

[Figure]

Figure 7: The difference of the predictive RMSE between the proposed method and other uncertainty processing approaches for precipitation forecasting.

**The section "results" is too brief. The authors may analyse how the proposed framework improve RMSE and uncertainties in detail, for example, the contribution of model uncertainty, input data uncertainty and target data uncertainty.**

**Response:** Thanks for your suggestions. The "results" section is expanded by including the contribution of model uncertainty and data uncertainty, the distribution of mode weights and the detailed analysis of how the proposed framework improves RMSE and uncertainties.

Figure 9 demonstrates the relationship between the predictive uncertainty by the proposed method and the summation of the predictive uncertainty from data and model. The predictive uncertainty by the proposed method ($\sigma_{xy\theta}$) agrees well with the summation of the predictive uncertainty from data and model, with a regressed slope of 1.004 and a $R^2$ of 0.98, indicating the good consistency of the predictive uncertainty between joint and separate uncertainty modeling. The median predictive uncertainty is 8.04, 1.40 and 9.49 mm for data, model and joint data-model uncertainty modeling, respectively in the precipitation forecasting experiments (Figure 10). The data and model uncertainty account for ~85% and ~15% of the total predictive uncertainty by the proposed method respectively. The model structure is fixed in this study and thus the model uncertainty comes from mainly the parameters. The dropout method enables the random abandoning of network parameters over the specific layer and prevents the overfitting of the forecasting model. The distributions of model weights are changed when incorporating the data uncertainty into the objective function (Figure 11). The model weights become larger and more dispersive relative to the prediction without uncertainty processing or with model uncertainty consideration. The data uncertainty is propagated into the prediction through model weights and thus the data uncertainty contributes more to the predictive uncertainty than model uncertainty. Although the model weights appear more scattered, the predictive accuracy exhibits small improvement due to the uncertainty consideration in the objective function.

[Figure]

Figure 9: The relationship between the summation of predictive uncertainty from data (input and target data) and model and the predictive uncertainty from joint consideration of data (input and target data) and model.

[Figure]

Figure 10: The distributions of the predictive uncertainty from data, model and joint data-model

modeling methods for precipitation forecasting.

Figure 7 demonstrates the difference of the predictive RMSE between other uncertainty processing approaches and the proposed method for precipitation forecasting. It can be seen that there is little difference of the predictive RMSE between the proposed method and other uncertainty processing approaches. However, the RMSE is larger for some uncertainty processing methods relative to the proposed method in some areas, including the Hainan and Taiwan provinces and part of eastern China. It is likely that the predictive RMSE is reduced in the proposed method by improving the prediction performance in the areas with accelerated water cycle and abundant rainfall. **The target data uncertainty is incorporated into the objective function to pay different attentions in space during the training process, leading to a different weight distribution of the forecasting model for the proposed method relative to others. The changing weight distribution improves the forecasting accuracy in the southeast areas by reducing the forecasting error of extreme precipitation.**

**Minor comments:**

**The abstract is not complete in my point of view. There are four parts in the abstract usually, i.e., background, method, results and conclusion. The results and conclusion are missing in the abstract.**

**Response:** Thank you for the instructive comments. The results and conclusion parts are added in the abstract in the revised version.

The experimental results indicate that the proposed joint uncertainty modeling framework for precipitation forecasting exhibits better forecasting accuracy (improve RMSE by 1-2% and R-square by 1-7% on average) relative to several existing methods, and could reduce the predictive uncertainty by ~28% relative to Loquercio et al. (2020)'s approach. The incorporation of data uncertainty in the objective function changes the distributions of model weights of the forecasting model and the proposed method can slightly smooth the model weights, leading to the reduction of predictive uncertainty relative to Loquercio et al. (2020)'s method. The predictive accuracy is improved in the proposed method by incorporating the target data uncertainty and reducing the

forecasting error of extreme precipitation. The developed joint uncertainty modeling method can be regarded as a general uncertainty modeling approach to estimate predictive uncertainty from data and model in forecasting applications.

**Line 150. Is it seasonal or weekly?**

**Response:** Thanks for pointing out this error. It's weekly.

**Line 156. NECP or NCEP?**

**Response:** It should be NCEP. Corrected in the revised version.

**Lines 268-269. This sentence is unclear; what is the point the authors want to make with it?**

**Response:** We would like to express that the sampling methods are used to sample from the data distribution to produce ensemble forecasts. The sampling process is conducted both for predictor data and predictand data. This is clarified in the revised version.

**There are many symbols without introduction in Figure 2, such as $x_n^l$, $\sigma_n^l$, etc. The $x_n^l$ is also seen in Equation 16 without introduction.**

**Response:** Sorry for the unclear clarification. The unclarified symbols are explained in the revised manuscript.

$\sigma_x$ and $\sigma_y$ are the predictor and predictand uncertainty, respectively. $x_n^l$ and $\sigma_n^l$ represents the propagated data value and uncertainty, respectively, for the $l$th network layer.

**Lines 319-320. Which deep learning network is used in this study? CNN, RNN, LSTM or all of them? Please make it clear.**

**Response:** The CNN network is used in this study. This is clarified in the revision.

**Line 376. Typographical error.**

**Response:** This error is corrected in revision.

**Lines 453-454. Please rephrase.**

**Response:** This sentence is rephrased as "The higher the data and model uncertainties, the more divergent and less reliable the forecasting".

**Anonymous Referee #2**

**Referee comment on "Quantifying the uncertainty of precipitation forecasting using probabilistic deep learning" by Lei Xu et al., Hydrol. Earth Syst. Sci. Discuss., https://doi.org/10.5194/hess-2021-432-RC2, 2022**

**This study attempts to improve the accuracy of precipitation forecasting by jointly considering multi-source data-model uncertainties in deep learning based modeling framework. A case study conducted in the southern and northern China showed that the developed modeling framework is effective to reduce the uncertainty in precipitation forecasting. In my opinion, this study is valuable and the methodology developed is based on rigorous mathematical formulas that is worthy of recognition. Some of my main comments are listed below**

**Response:** Thank you very much for the reviewing of our manuscript.

**Line 9-26. It is suggested that some summative results should be added to the Abstract.**

**Response:** Thanks for your suggestion. The key results are included in the abstract in the revision. The experimental results indicate that the proposed joint uncertainty modeling framework for precipitation forecasting exhibits better forecasting accuracy (improve RMSE by 1-2% and R-square by 1-7% on average) relative to several existing methods, and could reduce the predictive uncertainty by ~28% relative to Loquercio et al. (2020)'s approach. The incorporation of data uncertainty in the objective function changes the distributions of model weights of the forecasting model and the proposed method can slightly smooth the model weights, leading to the reduction of predictive uncertainty relative to Loquercio et al. (2020)'s method. The predictive accuracy is improved in the proposed method by incorporating the target data uncertainty and reducing the forecasting error of extreme precipitation. The developed joint uncertainty modeling method can be regarded as a general uncertainty modeling approach to estimate predictive uncertainty from data and model in forecasting applications.

**Line 138-159. Adding some key formulas about the TCH algorithm can facilitate the understanding of whole framework.**

**Response:** Thanks. The key formulas for TCH method are added in the manuscript.

**Some variables in equations need further explanation, such as the 'I' in Equation (1).**

**Response:** We checked all the equations and symbols to make sure that all the variables are explained clearly. The $I$ in Equation (1) is the identity matrix.

**Line 209. Typographical error. Should be 'estimated'**

**Response:** This error is corrected.

**Line 349-366. There are many experimental settings. It is suggested to explain them in bullet points, or use a clearer presentation.**

**Response:** Thanks for your suggestion. A summary table is used to clearly express the experimental settings.

Table 1. A summary of the experiment setup in this study. The √ and × symbol indicates that the specific factor is considered and omitted, respectively.

| Experiments | Uncertainty processing | | Model uncertainty |
| --- | --- | --- | --- |
| | Data uncertainty | | |
| | Input data (predictor) uncertainty | Target data (predictand) uncertainty | |
| Experiment 1 | × | × | × |
| Experiment 2 | √ | × | × |
| Experiment 3 | √ | √ | |
| Experiment 4 (Srivastava et al., 2014) | × | × | √ |
| Experiment 5 (Loquercio et al., 2020) | √ | × | √ |
| Experiment 6 (This study) | √ | √ | √ |

**Line 377. In Figure 5, it is recommended to plot the uncertainty estimation results of all datasets for visual comparison.**

**Response:** Sure. The uncertainty estimation for all datasets is plotted in the revision.

[Figure]

Figure 5: The data uncertainty calculated by the TCH method. The uncertainty distribution is plotted according to the uncertainty over all the pixels of the study area.

**Line 402: Add two numbers estimated by Loquercio et al. (2020)'s and Srivastava et al. (2014)'s methods for an intuitive comparison.**

**Response:** Thanks for your suggestion. The specific mathematical values are added in this sentence for intuitive comparison. The average predictive uncertainty (10.859) based on the proposed method is smaller than the Loquercio et al. (2020)'s method (15.232), and the predictive $R^2$ is higher (0.539) than the latter (0.523).

**The Results Section lacks some detailed analysis on how the developed method can improve the prediction accuracy.**

**Response:** We added detailed analysis on how the proposed framework can improve the prediction accuracy.

Figure 7 demonstrates the difference of the predictive RMSE between other uncertainty processing approaches and the proposed method for precipitation forecasting. It can be seen that there is little difference of the predictive RMSE between the proposed method and other uncertainty processing

approaches. However, the RMSE is larger for some uncertainty processing methods relative to the proposed method in some areas, including the Hainan and Taiwan provinces and part of eastern China. It is likely that the predictive RMSE is reduced in the proposed method by improving the prediction performance in the areas with accelerated water cycle and abundant rainfall. The target data uncertainty is incorporated into the objective function to pay different attentions in space during the training process, leading to a different weight distribution of the forecasting model for the proposed method relative to others. The changing weight distribution improves the forecasting accuracy in the southeast areas by reducing the forecasting error of extreme precipitation.

[Figure]

Figure 7: The difference of the predictive RMSE between the proposed method and other uncertainty processing approaches for precipitation forecasting.

Table 2. The accuracy of precipitation forecasting based on different uncertainty processing strategies. The best RMSE and the highest R-square are shown in bold for each column.

| Uncertainty processing | RMSE | Uncertainty | R-square |
|---|---|---|---|
| No uncertainty | 25.138 | - | 0.503 |
| Predictor uncertainty | 25.065 | 14.729 | 0.517 |
| Predictor and predictand uncertainties | 24.679 | 9.290 | 0.533 |

| | | | |
|---|---|---|---|
| Model uncertainty (Srivastava et al., 2014) | 25.156 | 1.570 | 0.506 |
| Data and model uncertainties (Loquercio et al., 2020) | 25.056 | 15.232 | 0.523 |
| Data and model uncertainties (This study) | **24.531** | 10.859 | **0.539** |

[Figure]

Figure 11: The mean and standard deviation of squared weights of the precipitation forecasting model.

The predictive uncertainty is closely related to the input data uncertainty and model weights. In Figure 2, the input data uncertainty is propagated into predictive uncertainty through model parameters, while the target data uncertainty is incorporated into the training process by the objective function. The incorporation of uncertainty processing changes the training process and the distributions of model weights (Figure 11). The last CNN layer (Layer 13) weights of the precipitation forecasting model are generally higher than the other layers, which may contribute largely to the predictive uncertainty by propagating the data uncertainty. The mean and the standard deviation of the squared weights of the last CNN layer for the Loquercio et al. (2020)'s method are higher than other uncertainty processing methods, suggesting an overall larger and more dispersive weight distribution for Loquercio et al. (2020)'s method than the others, which could partly explain the high uncertainty of the Loquercio et al. (2020)'s method in Table 2.

---

## Editor Decision (ED1)

Hydrol. Earth Syst. Sci. Discuss., referee comment RC1
https://doi.org/10.5194/hess-2021-432-RC1, 2021
**Comment on hess-2021-432**

Anonymous Referee #1
* * *
Referee comment on "Quantifying the uncertainty of precipitation forecasting using probabilistic deep learning" by Lei Xu et al., Hydrol. Earth Syst. Sci. Discuss., https://doi.org/10.5194/hess-2021-432-RC1, 2021
* * *
In the presented manuscript, a joint uncertainty modeling method is proposed. The input data uncertainty, target data uncertainty and model uncertainty are jointly modeled in a deep learning precipitation forecasting framework to estimate the predictive uncertainty. The results show that the proposed method can improve precipitiation forecasting accuracy and reduce predictive uncertainty. Having said that I am lost and confused. Here are some of my major concern regarding the presented study:

- Precipitation forecasting datasets are usually daily. In this study, the authors use three datasets are all daily. Why the authors convert the data to weekly data? In Line 134, the historical three consecutive weeks are used to forecast the precipitation in the target week. How to determine the "three" weeks?
- Why the authors use NCEP R2, ERA-5 and MERRA-2 data? For exsample, NCEP CFSv2 also have weekly precipitation forecasting data. In this study, MERRA-2 is used as the reference data. Do different reference data in the uncertainty estimation?
- How to determine the structure of the deep learning model in Figure 3? Besides the model parameters, the model structure can also generate uncertainty. Have the authors considered this part of uncertainty in this study?
- Lines 423-424: In the places where the annual rainfall is abundant, the water cycle process is accelerated and the precipitation observations may suffer from large uncertainty. Why this uncertainty dosen't exhibit in Figure 7? There are no larger uncertainty observed in the southern China in Figure 7. Is it contradictory?
- How to calculate RMSE and uncertainty in Table 1? Are they the average of all the grid cells? How dose the uncertainty processing to improve forecasting accuracy and reduce predictive uncertainty? Why some methods have considered uncertainty processing but their RMSE increase compared with the no-uncertainty method? Loquercio's method also considers the data and model uncertainties. Why the uncertainty of Loquercio's method is so large?
- The section "results" is too brief. The authors may analyse how the proposed framework improve RMSE and uncertainties in detail, for example, the contribution of model uncertainty, input data uncertainty and target data uncertainty.

Minor comments:

- The abstract is not complete in my point of view. There are four parts in the abstract usually, i.e., background, method, results and conclusion. The results and conclusion are missing in the abstract.
- Line 150. Is it seasonal or weekly?
- Line 156. NECP or NCEP?
- Lines 268-269. This sentence is unclear; what is the point the authors want to make with it?
- There are many symbols without introduction in Figure 2, such as $x_n^{(l)}$, $\sigma_n^{(l)}$, etc. The $x_n^{(l)}$ is also seen in Equation 16 without introduction.
- Lines 319-320. Which deep learning network is used in this study? CNN, RNN, LSTM or all of them? Please make it clear.
- Line 376. Typographical error.
- Lines 453-454. Please rephrase.

[Figure]

Hydrol. Earth Syst. Sci. Discuss., referee comment RC2
https://doi.org/10.5194/hess-2021-432-RC2, 2022
**Comment on hess-2021-432**

Anonymous Referee #2
* * *
Referee comment on "Quantifying the uncertainty of precipitation forecasting using probabilistic deep learning" by Lei Xu et al., Hydrol. Earth Syst. Sci. Discuss., https://doi.org/10.5194/hess-2021-432-RC2, 2022
* * *
This study attempts to improve the accuracy of precipitation forecasting by jointly considering multi-source data-model uncertainties in deep learning based modeling framework. A case study conducted in the southern and northern China showed that the developed modeling framework is effective to reduce the uncertainty in precipitation forecasting. In my opinion, this study is valuable and the methodology developed is based on rigorous mathematical formulas that is worthy of recognition. Some of my main comments are listed below:

- Line 9-26. It is suggested that some summative results should be added to the Abstract.
- Line 138-159. Adding some key formulas about the TCH algorithm can facilitate the understanding of whole framework.
- Some variables in equations need further explanation, such as the 'I' in Equation (1).
- Line 209. Typographical error. Should be 'estimated'
- Line 349-366. There are many experimental settings. It is suggested to explain them in bullet points, or use a clearer presentation.
- Line 377. In Figure 5, it is recommended to plot the uncertainty estimation results of all datasets for visual comparison.
- Line 402: Add two numbers estimated by Loquercio et al. (2020)'s and Srivastava et al. (2014)'s methods for an intuitive comparison.
- The Results Section lacks some detailed analysis on how the developed method can improve the prediction accuracy.